# A Hybrid Methodology to Assess Cyber Resilience of IoT in Energy Management and Connected Sites

**DOI:** 10.3390/s23218720

**Published:** 2023-10-25

**Authors:** Amjad Mehmood, Gregory Epiphaniou, Carsten Maple, Nikolaos Ersotelos, Richard Wiseman

**Affiliations:** 1Secure Cyber Systems Research Group (CSCRG), WMG, University of Warwick, Coventry CV4 7AL, UK; dramjad.mehmood@warwick.ac.uk (A.M.); gregory.epiphaniou@warwick.ac.uk (G.E.); cm@warwick.ac.uk (C.M.); 2Institute of Computing, Kohat University of Science & Technology, Kohat 46000, Pakistan; 3Department of Computer Science and Creative Technologies, University of the West of England, Bristol BS16 1QY, UK; 4BT Group, 5th Floor, Orion Building, Adastral Park, Martlesham Heath, Ipswich IP5 3RE, UK; richard.wiseman@bt.com

**Keywords:** cyber resilient model, blockchain, digital twins, critical national infrastructure (CNI), critical success factor (CSF), key result areas (KRAs), key performance indicators (KPIs), safety

## Abstract

Cyber threats and vulnerabilities present an increasing risk to the safe and frictionless execution of business operations. Bad actors (“hackers”), including state actors, are increasingly targeting the operational technologies (OTs) and industrial control systems (ICSs) used to protect critical national infrastructure (CNI). Minimisations of cyber risk, attack surfaces, data immutability, and interoperability of IoT are some of the main challenges of today’s CNI. Cyber security risk assessment is one of the basic and most important activities to identify and quantify cyber security threats and vulnerabilities. This research presents a novel i-TRACE security-by-design CNI methodology that encompasses CNI key performance indicators (KPIs) and metrics to combat the growing vicarious nature of remote, well-planned, and well-executed cyber-attacks against CNI, as recently exemplified in the current Ukraine conflict (2014–present) on both sides. The proposed methodology offers a hybrid method that specifically identifies the steps required (typically undertaken by those responsible for detecting, deterring, and disrupting cyber attacks on CNI). Furthermore, we present a novel, advanced, and resilient approach that leverages digital twins and distributed ledger technologies for our chosen i-TRACE use cases of energy management and connected sites. The key steps required to achieve the desired level of interoperability and immutability of data are identified, thereby reducing the risk of CNI-specific cyber attacks and minimising the attack vectors and surfaces. Hence, this research aims to provide an extra level of safety for CNI and OT human operatives, i.e., those tasked with and responsible for detecting, deterring, disrupting, and mitigating these cyber-attacks. Our evaluations and comparisons clearly demonstrate that i-TRACE has significant intrinsic advantages compared to existing “state-of-the-art” mechanisms.

## 1. Introduction

Whilst businesses today are ever increasingly reliant on technology than before, and technological mediators underpin almost every critical civil society function, vulnerabilities exist within technological mediators, and these vulnerabilities have the potential to be exploited by adversaries, hence directly impacting the execution of business operations. Cyber security offers potentially valuable insights to enable security-related risks to be identified, quantified, assessed, and showcased to nontechnical C-Suite decision makers and budget holders. Hence, successful security management enables informed organisational decision making, improves cyber security strategy, and connects with the organisation’s needs and risk appetite, allowing it to achieve its long-term objectives more effectively.

Critical national infrastructure (CNI) comprises the essential and critical assets, such as information technology (IT), networks, facilities, etc., that underpin the provision of food, energy, health, emergency services, technology, transport services, and the interrelated processes that provide day-to-day essential services. CNI impacts individual life and a nation-state’s overall economic growth. The nature of CNI itself and its vital role in provisioning vital real-time, low-latency e-utilities means that standard security solutions, frameworks ISO standards, incident response and forensics are insufficient to adequately protect mission-critical connected CNI and OP facilities, systems, and sites from damage or loss. This research followed the generic ISO standard ISO31000 [1] for risk monitoring and risk communication and the CVSS scoring system. Organisations supporting CNI had just 36% of some 370 participating entities and already had sufficient cyber resilience. Siemens and the Ponemon Institute explored 64% of sophisticated attacks against key utilities. Keeping up with the industrial cyber threats sector was rated as a top challenge, and around 54% of respondents fully expected that an attack on CNI would occur in the next year [2]. According to the same, only 35% of survey participants reported that they have an IoT security strategy in place (of which only 28% said that they had implemented it).

Similarly, another survey [3] found that 80% of organisations had experienced cyber attacks on their IoT devices in the past year. However, refs. [2,3] found that 26% of the organisations did not use IoT-specific security protection technologies. These surveys demonstrate the inherent security limitations of many IoT devices (many have “lite” weak onboard, built-in security features), hence the urgent need for organisations to move at pace proactively to invest in IoT cyber security. Despite weak security measures, existing risk assessment methods are inappropriate for low-latency dynamic OP systems such as IoT devices. Hence, an extensive and dynamic cyber risk assessment method needs time to cope with the requirements of a resilient IoT system.

The i-TRACE project is a collaboration between the University of Warwick, British Telecommunications plc (BT) the national leader in network operations and management, Cisco the world leader in network routing equipment manufacturing, and Senseon a UK-based medium-sized enterprise. Through a developed system from Senseon, based on AI and threat data recovery, wrapped around network threat modelling and knowledge from the University of Warwick, the system can provide an enhanced discoverability system [4] that cannot be matched with the mitigation algorithms. Alongside discovery, i-TRACE provides a resilient trust system based on a blockchain signature system and Cisco’s Assured Transport System. 

The i-TRACE project’s key performance indicator (KPI) assessment is a realistic, measurable, secure, low-cost, and long-life IoT cyber security solution that leverages existing edge device technologies in connection with distributed blockchain technology to add immutable identity, time, and content metadata to data in motion. The functions of KPIs within i-TRACE drive continuous optimisation, distinguishing between what has been implemented correctly and which areas still need attention and facilitating continuous fine-tuning of the system and controls. Since security threats constantly evolve, security management is a constant process, reliant upon KPIs to measure performance and derive security decisions as required. Since info-sec needs to be considered primarily a key managerial concern instead of a purely technological issue, KPIs are necessary to evaluate the success of particular software engineering activities, lifecycles, devices, third-party supplier products, networks, and architectures. Thus, KPIs need always be aligned with the objective and goals. This approach would be through key result areas (KRAs), critical success factors (CSFs), or key drivers of success [5].

The i-TRACE project KPI assessments emphasise the significance of security concerns by revealing the impacts that these have on the following use cases: connected and energy management sites. i-TRACE endorses the fact that cyber security concerns can potentially restructure the use cases and enhance a system’s overall efficacy and efficiency. Figure 1 below shows an architectural workflow diagram comprising different end-points. The solution is installed on the Preston site. It is used to collect data from each of these devices, regardless of the communication protocols, while guaranteeing the data’s integrity and interoperability. Potential benefits include securing and better managing the construction sites and accelerating the deployment of IoT sensing capabilities into more construction sites, including, but not limited to, drivers, location, actors, and success criteria. Hitherto, such sites have been conservatively managed/deployed. After selecting the project management KPIs, it is essential to define them in such a way as to clarify, articulate and support the goals of the project. The most important aspect of a KPI is to be “S.M.A.R.T.E.R”. (specific, measurable, attainable, realistic, time-bound, evaluation, and re-evaluation) for project success. Such KPIs not only help to ensure that the project is directed toward the right direction, but if the project deviates from its predefined success path, KPIs help to rectify its forward trajectory [6,7,8,9].

The i-TRACE KPI-based methodology aims to address, hence remediate, the challenges of securing prevalent IoT devices. It seeks to offer reliable, low-cost and long-life IoT solutions in the context of various heterogeneous edge IoT devices’ deployments that leverage distributed blockchain technologies [10]. In the past, energy management approaches employed costly to monitor and energy inefficient solutions. However, due to technological advancements in IoT, numerous bespoke, end-to-end, cost-effective, and efficient systems have been deployed and, hence, are readily available. However, many so-called “low-cost” IoT devices and the end-to-end solutions have not been hitherto designed with cyber resilience in mind. This inevitably makes such bespoke “solutions” insecure and vulnerable devices with penetration points (attack vectors); as a result, an unauthorised user or attacker can take advantage of these attack vectors to penetrate and change or hack important data [11]. Generically, deploying such devices within end-to-end solutions creates a “trust” deficit. This deficit can fundamentally undermine the confidentiality, availability, and integrity of mission-critical CPNI systems, as exemplified by our two chosen use cases described in the following sections.

The key contributions of this paper are the following:A novel i-TRACE KPI assessment methodology is proposed to overcome the interoperability issues and reduce the cyber risks in IoT systems, using the use case of energy management and connected sites.The proposed resilient methodology leverages the digital twins’ modern technology and distributed ledger technologies.The security controls and vulnerability management of IoT devices are demonstrated.The SMARTER KPIs are followed to embody a set of wide-ranging countermeasures to the cyber security challenge of IoT.

The rest of the paper is organised as follows. Section 2 discusses the literature review. Section 3 presents the i-TRACE KPIs assessment methodology. The KPIs assessment, performance measurement, and risk evaluation are performed using two use cases, the connected sites and energy management sites, in Section 4. The conclusion of the work is presented in Section 5.

## 2. Literature Review

Numerous previous authors have sought to address the security challenges posed by IoT-enabled CPNI (Centre for the Protection of National Infrastructure) systems. In [12], the authors presented an approach to asset identification using a multicriteria-based decision theory to overcome the challenges of identifying critical assets of critical infrastructures (CIs). Whilst a valuable contribution to the literature, the authors do not offer a method for making a critical decision. A novel structured risk management approach was presented in [13], wherein the authors proposed specific techniques specifically designed to mitigate the hazardous events of internal and external impacts of a given CI. Their research followed the generic ISO standard ISO31000 for risk monitoring and communication. Interdependencies were also discussed within [12]. However, the authors offered no guidelines for calculating risk levels and their mitigation. In [14], the authors presented an overview of cyber-critical assets within CI. Strategic planning for civil protection and risk management activities was offered; however, the key issues of both threat impact and prevention were not explicitly addressed. In [15], the authors described and applied the UML (Unified Modelling Language) in telecommunication systems by adopting a model named TVRA (Threat, Vulnerability, and Risk Analysis). The TVRA, based on UML-centric modelling, enabled the authors to articulate and systematically analyse the system’s security objectives, weaknesses, assets, vulnerabilities, threats, and detrimental incidents. In [16], Clarizia et al. presented a multilevel graph methodology that collects and analyses sensor data using context dimension trees, Bayesian belief networks, and ontologies to support decision-making. In [17], Wang and Liu proposed a novel attribute, “location”, and presented a detailed vulnerability analysis for the Multimedia Subsystem (IMS) network and Internet Protocol (IP), designed to identify the weaknesses of IMS systems. However, many critical assets were not identified in their model. In [18], the authors proposed a novel model to determine the vulnerabilities that arise via unexpected interactions between system components. All the components were modelled using a high-level specification language to capture all possible behaviours of a system. Each behaviour was further analysed using an automated verification technique to identify security-related violation(s). Ezell [19] proposed a model that quantifies vulnerabilities using the IVAM (Infrastructure Vulnerability Assessment Model), applying the model to a medium-sized system. This paper did not specifically identify the overall assets; instead, the aim was to quantify the system’s security vulnerabilities fully. In [20], the authors reviewed the “state-of-the-art” cyber security risk assessment methodologies commonly used in SCADA (supervisory control and data acquisition) system design and deployment. Various risk assessment techniques were examined and analysed, including stages of risk management, application domain impact measurement, risk management, and probabilistic data evaluation tools. McQueen et al. [21] presented a technique to estimate the time needed for an attacker to compromise a system. This model estimates the expected value of known and visible vulnerabilities as well as the skill level of the attacker. The model was used to assess the risk reduction in a SCADA system. The authors also presented a method to estimate the time to compromise. They used the standard of the North American Electric Reliability Corporation (NERC), i.e., CIP-002 through CIP 009, to provide a security framework that supports the reliable operation and maintenance of electric power grids. In [22], the authors proposed a risk reduction model on a partial SCADA system. Their methodology was developed by estimating quantitative risk reduction using a graph-theoretical approach. Both cloud and blockchain were leveraged.

However, in all the preceding studies, the key issue of inefficient collaborations among project participants, which helps complete projects on time and within budget, remains unaddressed. Several studies [23,24,25] showed that DT technology has the immense potential to support information sharing among project participants. DTs are the virtual representation of digital assets using sensor data to represent real-time information visually. To share accountable information among fragmented participants using digital twins (for example), all the transactions need to be transparent without any potential for adversarial or other (malicious) manipulation. The shared data needs to be tamper-proof. However, heterogeneous DT issues in data management, including data storage, security, and sharing, have yet to be thoroughly realised.

DTs using a hybrid approach can selectively store and share important i-TRACE information traceability. The hybrid approach adds authentication and traceability to any transaction shared amongst participants. Decentralised mechanisms authenticate and attest to the accuracy and integrity of transactions amongst project participants. These serve to facilitate, verify, or otherwise automatically enforce agreement terms embedded within contracts. Consequently, collaboration is supported whilst at the same time reducing unnecessary interactions among participants (i.e., improved project efficiency and customer satisfaction; see [24]). Table 1 below aims to show that constructing an entirely new architecture is not required. Instead, our contribution leverages the “best” (optimal) features and functionality of existing architectures in a hybrid manner [25].

It is clear that there is a genuine need to identify the critical assets, vulnerabilities, and threats to CPNI systems. It has been observed that there is currently a lack of systematic approach to support the critical national infrastructure (CNI) organisations via identifying their critical assets; hence, cyber security vulnerabilities and threats. Furthermore, what is needed is a systematic KPI-driven method of asset identification and vulnerability assessment consolidated with the effect of the vulnerabilities identified upon cyber threats, hence, associated risk. In the case of our novel i-TRACE solution (which leverages a private blockchain), only known participants are admitted to the network, thus confirming that only fully authenticated and authorised nodes can mine and append new blocks. The justification and choice of blockchain as the security architecture central to i-TRACE is that it drastically reduces the possibility of the injection of malicious nodes and/or other adversarial interference, even by a nation-state actor. This does not mean that blockchain can be made totally secure or that the data are immutable, per se [25]. However, due to numerous inherent advantages, it is an order of magnitude more “secure” than conventional choices such as private cloud.

## 3. i-TRACE KPIs Assessment Methodology

The i-TRACE KPI project has a set of proposed key results areas (KRAs) or key performance areas (KPAs) for each use case. Different parameters of each use case are measured accordingly to determine the impact of our innovation on the KPIs established in the programme. A dedicated methodology is also presented herein to assess the KPIs for the i-TRACE project out of the KPAs. Using that methodology, the key results areas (KRAs) for both use cases will be assessed and decomposed into smaller, more specific, quantifiable, and measurable indicators. This means selecting SMARTER (specific, measurable, attainable, result-oriented, time-based, evaluated, and re-evaluated) indicators. Data will be gathered in the data collection phase, during which each hand will be analysed using characteristics, such as its name, description, objectives, type (quantitative, qualitative), effort (low, medium, high), metric setup (scale, formula, range, weight, percentage), unit, assessment method, possible tool, analysis frequency, comments, etc., in the analysis and design phase. KPI questions will be framed to develop better and more meaningful performance indicators in order to validate the alignment of the goals set and achieved.

### 3.1. KRA of i-TRACE

As already discussed in the introduction section, key result areas (KRAs) or key performance areas (KPAs) are established in order to evaluate the effects of our innovation on the KPIs. A dedicated methodology has also been introduced to evaluate the KPIs for the project. As shown in Figure 2, the KRAs help define the scope and the optimum outcomes and results. To succeed, critical items require long discussions between consortium members to go through the pros and cons of the UCs individually; the following list of KRAs has been filtered for both the use cases (UCs) of i-TRACE.

#### 3.1.1. UC-1: Connected Sites (CoS) KRAs

Level of interoperability and immutability aspects achieved.Level of reduction in cyber security risks.

Level of safety for people on construction sites.

#### 3.1.2. UC-2: Energy Management Sites (EMS) KRAs

Level of reduction in cyber security risks (UC-1).Level of data immutability achieved (UC-1).Reduction in the attack surface.Level of minimisation of attack impacts.

### 3.2. Decomposition of KRAs

According to the methodology set out in Figure 2, after having KRAs in hand, they are further broken down into smart, smaller, more specific, practical, quantifiable, and measurable parts to achieve the importance of the meaning of each KRA regarding use cases. Therefore, the following decompositions were performed from Level I to Level III to gain insight into each use case.

#### Level I: Decomposition of KRAs of CoS and EMS

The following are explanations of the KRAs related to CoS and EMS use cases:Level of interoperability and immutability aspects achievedLet *D* = (*V*, *E*), where the vertex set *V* = {*v*1, vs. 2,⋯vs. *n*} is the set of *n* systems supporting the operational thread, and the edge set *E* = {*e*1, *e*2,⋯*enn*} is the set of directed connections between systems (including loops). Define the spin matrix *S* = [*sij*], *sij* ∈ −1, 0, +1, *i*, *j* = 1⋯*n* as a modified adjacency matrix and the multiplicity matrix *C* = [[*cij*]*n* × *n*, *cij* ∈ ≥0]*I*, *j* = 1⋯*n* as a spin matrix multiplication, where *Cij* is the number of times a system pair is repeated when the elements of T are taken two at a time in a forward direction. *M* = [*Cij* × *sij*]*n* × *n*, where *M* is defined as the interoperability matrix [26]. The data (or metadata) are securely distributed across several entities, ensuring integrity and lowering the risk of loss whilst offering an audit trail (in the case of a malicious actor). The “append-only” model inherent to blockchain provides all participants of the private blockchain with full transparency viz a viz activity. Enabling both a “holistic” viewpoint and forensic analysis to be performed for a “deeper dive” as desired [27].Level of reduction in cyber security risksRisk management is mission-critical to all business functions, and as companies grow, it becomes an ever more complex task. Managing risk at strategic and operational levels requires the nuanced consideration and evaluation of inherent trade-offs. Eliminating all risks, including security and technological risks, through assurance activities is simply impractical in terms of cost–benefit analysis. The intrinsic “tug-of-war” between productivity and security is tricky to manage; many risk professionals embedded within complex organisations are simply managers outmanoeuvred by day-to-day operational demands. The need to provide heterogeneous stakeholders within a large-scale healthcare provider access to Big Data inevitably means that such systems expose themselves to internal and external threat actors. Therefore, business risk managers should pragmatically conceptualise generic safety, cyber risk, and cyber risk mitigation within the context of CPNI.Level of safety for people on construction sitesWorking on construction sites is a hazardous activity with a high risk of on-site accidents, off-site hazards, health issues, and safety risks. The best ways to avoid construction site hazards will place you and your building sites in the optimal position to continue to attract the best workers. Injury, illnesses, mental health, and long-term damage are some of the main negative outcomes. Some of the main causes of these accidents are lack of communication/unclear training, electrocution, unsafe access/egress, unsafe spoil-pile placement, and lack of protective systems in place. In recent years, there has been a realisation that the reliability of complex work systems in achieving organisational goals safely depends on the work structures and the technical arrangements that the perceived level of risk and safety, the accident rate, the level of employees’ cooperation, the safety attitude of managers and employees, the level of employees’ physical risk in a workplace, and level of safety information indicate as key safety parameters [27].Reduction in the attack surfaceOne of the key ways to assess the vulnerability of a system is to assess and measure the number of ways an application, system, etc., can be exploited. The attack surface consists of a compendium of vulnerabilities an attacker could exploit to compromise the network system, device, or API. The larger the system’s attack surface is, the more vulnerable the system is to attacks and the more damage that is likely to result from the attacks [28]. By reducing the attack surface, we can protect the devices and networks of i-TRACE use cases, as it leaves hackers with fewer ways to perform their attacks. A large attack surface provides attackers with multiple points to gain illegal access to sensitive data such as personally identifiable information of employees and customers, financial transaction records, sensitive information exchange, and more. Continuous attack-surface review is needed to keep pace with technological and platform protocol evolution.Reduction in attack vectorsAttack vectors are potential points the attackers can use to penetrate the IoT environment by exploiting the vulnerabilities of both data and network. Each point, such as protocols, access points, and services, represents a vulnerability. To identify the attack vectors, it is essential to clearly understand the IoT environment and the most common devices used in each IoT domain [10].

### 3.3. Use Case(s) Field Data Collection

Sustainability performance management collects data in two ways: 1. Automatic data collection, which refers to collecting KPIs via automatic scripts, which access the corresponding systems to gather the data; and 2.Manual data collection refers to collecting KPI data via correspondence with users who provide answers manually. Typical data collection methods include surveys, questionnaires, interviews, sensor data collection, focus groups, automated machine data collection, and collection of archival data [29].

#### Data Collection for Energy Management and Connected Sites

In this section, data related to each KPI are collected and stored appropriately. Afterwards, an analysis will be performed to determine the results of each KPI accordingly.

### 3.4. Analysis and Design Indicators

The criteria used in the method, along with their definitions, can be found in Table 2. This list is a working subset of the original twenty criteria previously identified by Horst and Weiss in 2015 [30]. Each criterion is ranked numerically in descending order by each stakeholder, using a rank sum method. For example, a one is assigned to the most important criterion, a two is assigned to the second most important criterion, and so forth.

The key performance indicator analysis for connected and energy management sites for each of the key KPIs for both our use cases, the corresponding KPI name, explanation, unit, formula, relevance, and time required to track each KPI [31] are presented below in Table 3.

### 3.5. KPQs for Connected Sites and Energy Management Sites

KPQs (key performance questions) help companies develop better, more meaningful, and useful performance indicators. This section presents the KPQs related to each finalised KPI for both the use cases: connected sites and energy management. KPQs help to optimise the tracking of the business’s goals and to indicate if the system is heading in the right direction.

### 3.6. Action Plan and Reporting

An audience and access to the KPI define the primary audience of the KPI, i.e., who these data are for and who will have access to them. The key performance indicator should always include an expiry date or revision date.

A smart dashboard will be designed to measure and report each indicator. The dashboard will be designed to perform all the designed tasks well before the set time and generate different types of alerts to guide its end users to take appropriate steps accordingly. A visual display (VD) highlights the most important information to assist in decision-making and performance management. The reporting frequency coordinates the data collection and ensures that the data are current and up to date. Performance management will be carried out autonomously.

## 4. KPIs Assessment: Connected Sites and Energy Management Sites

This section demonstrates an assessment of each KPI associated with the use cases (connected sites and energy management sites) of the i-TRACE project, filtered out in the section above.

### 4.1. Interoperability of the Use Cases

The i-Score methodology is used to calculate the interoperability of both the use cases of i-TRACE. The methodology is firmly based on the concepts of an operational thread and an interoperability spin. An operational thread is defined as a sequence of activities where each activity is supported by exactly one system (mechanism). An interoperability spin is defined as an intrinsic property of a system pair, which indicates the quality of the pair’s interoperation. Borrowing from physics, spin is a quantised intrinsic property. In this report, based on the i-TRACE’s use cases, i.e., energy management and construction sites, the word spin is used in connotation to describe the intrinsic interoperability between two devices, *i*, *j*, and quantise it as *sij* ∈ −1, 0, +1 (Table 4). To this end, the best spin (+1) is assigned when two devices can communicate without any translation (human or machine). An example of a system pair with *sij* = +1 is a sensor and a gateway. The next best spin (0) is assigned to a device pair, which requires an intervening device (nonhuman) to perform a machine translation to allow them to interoperate. An example of a device pair with *sij* = 0 is two devices or sensors that require gateways to interoperate. The worst spin (−1) is assigned when the only way for two devices or sensors to interoperate is if a human system intervenes and translates. A *sij* = −1 spin is often assigned between two human systems when they require a third human to perform language translation services in order for them to communicate, conduct business, or otherwise interoperate [32].



T={1,2,2,2,3,3,3,4,5,5,5,6,7,8}.



**A** = {(1,2), (1,2), (1,2), (1,3), (1,3), (1,3), (1,4), (1, 5), (1,5), (1,5), (1,6), (1,7), (1,8).

   (2,2), (2,2), (2,3), (2,3), (2,3), (2,4), (2,5), (2,5), (2,5), (2,6), (2,7), (2,8)

   (2,2), (2,3), (2,3), (2,3), (2,4), (2,5), (2,5), (2,5), (2,6), (2,7), (2,8)

   (2,3), (2,3), (2,3), (2,4), (2,5), (2,5), (2,5), (2,6), (2,7), (2,8)

   (3,3), (3,3), (3,4), (3,5), (3,5), (3,5), (3,6), (3,7), (3,8)

   (3,3), (3,4), (3,5), (3,5), (3,5), (3,6), (3,7), (3,8)

   (3,4), (3,5), (3,5), (3,5), (3,6), (3,7),(3,8)

   (4,5), (4,5), (4,5), (4,6), (4,7), (4,8)

   (5,5), (5,5), (5,6), (5,7), (5,8)

   (5,5), (5,6), (5,7), (5,8)

   (5,6), (5,7), (5,8)

   (6,7), (6,8)

   (7,8)}

   (1, 2) = 3, (1, 3) = 3, (1, 4) = 1, (1, 5) = 3, (1, 6) = 1, (1, 7) = 1, (1, 8) = 1

   (2, 2) = 3, (2, 3) = 8, (2, 4) = 3, (2, 5) = 8, (2, 6) = 3, (2, 7) = 3, (2, 8) = 3

   (3, 3) = 3, (3, 4) = 3, (3, 5) = 9, (3, 6) = 3, (3, 7) = 3, (3, 8) = 3

   (4, 5) = 3, (4, 6) = 1, (4, 7) = 1, (5, 8) = 3

   (5, 5) = 3, (5, 6) = 3, (5, 7) = 3, (5, 8) = 3

   (6, 7) = 1, (6, 8) = 1

   (7, 8) = 1

$C=\begin{array}{cccccccc} 0 & 3 & 3 & 1 & 3 & 1 & 1 & 1\\ 0 & 3 & 8 & 3 & 8 & 3 & 3 & 3\\ 0 & 0 & 3 & 3 & 9 & 3 & 3 & 3\\ 0 & 0 & 0 & 0 & 3 & 1 & 1 & 3\\ 0 & 0 & 0 & 0 & 3 & 3 & 3 & 3\\ 0 & 0 & 0 & 0 & 0 & 0 & 0 & 1\\ 0 & 0 & 0 & 0 & 0 & 0 & 0 & 1\\ 0 & 0 & 0 & 0 & 0 & 0 & 0 & 0 \end{array}$ $Spine=\begin{array}{cccccccc} 1 & 1 & 0 & 0 & 0 & 0 & 0 & 0\\ 1 & 1 & 1 & 1 & 0 & 0 & 0 & 0\\ 0 & 1 & 1 & 0 & 1 & 0 & 0 & 0\\ 0 & 1 & -1 & -1 & -1 & -1 & -1 & -1\\ 0 & 0 & 1 & -1 & 0 & 1 & 1 & 1\\ 0 & 0 & 1 & -1 & 1 & 0 & 0 & 0\\ 0 & 0 & 0 & -1 & 0 & 0 & 0 & -1\\ 0 & 0 & 0 & -1 & 0 & 0 & 0 & -1 \end{array}$



$M=C\;*\;Spine\begin{array}{cccccccc} 0 & 3 & 0 & 0 & 0 & 0 & 0 & 0\\ 1 & 1 & 1 & 1 & 0 & 0 & 0 & 0\\ 0 & 3 & 8 & 3 & 0 & 0 & 0 & 0\\ 0 & 0 & 3 & 0 & 9 & 3 & 0 & 0\\ 0 & 0 & 0 & 0 & -3 & -1 & -1 & -3\\ 1 & 0 & 0 & 0 & 3 & 0 & 3 & 3\\ 0 & 0 & 0 & 0 & 0 & 0 & 0 & 0\\ 0 & 0 & 0 & 0 & 0 & 0 & 0 & 0 \end{array}$




**I = 41-8 = 34**


**Total interactions:** 64

**Direct Communication**: Interoperable: (DCom): 21

**In-direct Communication**: (ICom): 33 

**Communication not possible**: CNP: 10

**DCom + ICom** = 21 + 33 = 34 => I

Hence it is proved that both the use cases are fully interoperable.

### 4.2. Level of Immutability Achieved for Both the Uses Cases

A hash of device data is written to the blockchain to ensure privacy instead of the data themselves. However, because of the potential volume of data, creating a separate transaction for every sensor update or even every device update might become prohibitively processor- and storage-intensive. Thus, sensors belonging to a device are grouped, and a number of updates are collated before creating a hash. For these data to be used for verification, other metadata needs to be stored along with the hash, namely, the device ID, the timestamp of the first and last updates included in the hash, and a count of the number of updates included in the hash (to allow further checking). When a transaction is verified, this metadata is used to extract data from the data exchange so that the hash can be regenerated and checked against the hash stored in the blockchain.

The diagram in Figure 3 shows the overall process. Devices’ updates are sent to the router and forwarded to the data exchange, where they are stored immediately and to a buffer for the blockchain. When the buffer contains a certain number of updates, a hash is generated and stored along with the metadata as data in a transaction. The hash must be generated precisely since even one character difference would produce a completely different hash. The approach used is as follows:Ensure the updates in the buffer are in time sequence.Create a data string for each update by using semicolons to join the individual data streams’ values (each data stream holds values for one of the device’s sensors).Join the data strings using semicolons to produce a combined data string containing all data for the update (note that in this and the previous step, no delimiter is actually required when joining values together, and the size of the hash is constant irrespective of the length of the input string; however, to simplify debugging and analysis, a human-readable delimiter is used).Compute the Keccak 256 hash of the combined data string.

It is conceivable that a device’s data might not all arrive in time sequence, so for robustness, the buffer used to hold data destined for the blockchain is actually three times the size of the number of updates used to generate a hash. When the buffer becomes full, it is sorted to ensure that out-of-sequence data are in the correct position, and then the oldest updates are removed and used to generate the hash. This is visualised and explained in Figure 4.

With reference to the “Verification Phase” block in the two figures, when verifying data, this is performed one transaction at a time, and the process is as follows:Obtain the data from the transaction. This is the device ID, the first and last timestamps of data included in the hash, the hash itself, and a count of the number of data groups used to create the hash (to enable a simple check).Use the device ID to map to the feed ID in the data exchange.Use the first and last timestamps to extract only the data relating to the transaction (and, therefore, the transaction’s hash) from the data exchange feed.Optionally check that the number of data points returned from the data exchange is the same as the expected number; if it is not, then the hash will not match, and there is no need for it even to be calculated.Create the hash using the data from the data exchange feed.Compare the newly-computed hash with that from the blockchain; if they match, then the data from the data exchange have not been altered and are the same as those sent originally (note that if the hashes do not match, this means that least part of the data do not match: this could be due to data that have been added to the data exchange feed in the transaction’s time range, data that have been removed from the feed, or data that have been altered in the feed. Whilst most or all of the data from the data exchange feed could be correct and unaltered, it is impossible to know what has changed, so all the data relating to the transaction must be treated as unreliable).

i-TRACE is a blockchain based on a decentralised P2P network and integrated with cryptographic processes. It can offer many new features and improve existing functionalities of IoT systems. Its security structure is more scalable than traditional ones, providing strong protection against data tampering, thus making it an attractive approach for addressing many security and trust challenges in large-scale IoT systems [33]. More specifically, the i-TRACE approach i. can be used to trace the measurements of IoT devices and prevent forging or modifying data. ii. can exchange data, establishing trust among themselves instead of going through a third party, significantly reducing the deployment and operation costs of IoT applications. iii. can eliminate a single source of failure within the IoT ecosystem, protecting the IoT devices used in both use cases from tampering. iv. can enable device autonomy via smart contracts, individual identity, and data integrity and can support P2P communication by removing technical bottlenecks and inefficiencies. v. although the configuration of IoT devices can be complex, the blockchain can be well adapted to provide IoT device identification, authentication, and seamless secure data transfer.

One of the most significant challenges in IoT scenarios is the vast amount of IoT data that is generated in a short period, and both the data hash and the data themselves need to be stored. If it does so, its immutability value would be 1; otherwise, it will be 0. Let I be the immutability vector, where the features of *I* = *i*0, *i*1,…, In, where n is the length of the I immutability vector, and each i-subsystem variable inside vector I represents a unique subsystem of immutability.

Consider the following scenario: a, b, c, d,…, *n* − 1, n are transactions; there are approximately 2000 transactions for a block of size I MB, with each transaction of 500 bytes between gateway and blockchain, all executed on the same block. Immutability or irreversibility is proved using the Merkle tree, a data structure constructed by recursively hashing pairs of transactions until only one hash, called the Merkle root, is used in Bitcoin to summarise all the transactions in a block. The following are the hash values of the transactions: h (a), h (b), h (c), h (d). The hashes are paired together, double-SHA256, resulting in Hash AB and Hash CD [34,35].

The diagram below (Figure 5) shows that the immutability in the communication or transactions between blockchain and other concerned devices will be maintained to 1 because if someone tries to pollute any block, then, of course, the entire block Merkle root’s hash code will be changed and the next block will be able to fetch its parent. The polluted node breaks from the chain and can be detected easily. Hence, the chain is almost immutable at different levels, as shown in Table 5.

### 4.3. Reduction in Cyber Security Risks of Both the Uses Cases

NIST conducts the risk assessment process, including risk (a) acceptance, if it is under a harmless level (risk appetite), (b) mitigation, by applying security measures, (c) transfer, or (d) avoidance, by removing the affected asset itself. This section will summarise the vulnerabilities of the IoT devices, which in the case of i-TRACE are resource-constraint sensor nodes and smart devices (Table 6). Based on a vulnerabilities assessment for the devices of the use case 1, compromise of the CIA (confidentiality, integrity and availability) triad is possible if: (a) the network services are not secure enough on the devices; (b) the gateways, digital twins, blockchain, and databases are not secured; (c) the device firmware is not validated; (d) insecure OS platforms or components from a compromised supply chain are used; and (e) hardening i.e. the process of securing a system by reducing its surface of vulnerability is not performed. Other vulnerabilities are related with (a) possibilities of cross-site scripting (XSS) attacks in Web applications, (b) possibilities of file directory traversal in the cloud server, (c) unsigned device updates, and (d) devices that ignore server certificate validity. Indeed, IOT suppliers shall use a Web application firewall to protect servers from HTTP traffic at the application layer. Recently, tremendous botnet-powered distributed denial of service (DDoS) attacks have exploited vulnerabilities of a few thousand IoT gadgets, utilising them to send bad traffic to valid websites. Vulnerabilities drastically increase the risks born due to the IoT devices, thereby mandating the need for a structured risk assessment process within the risk assessment frameworks. Table 7 depicts how the ranking of IoT risks can be calculated, resulting in five risk levels. If the risk ranks ≤ 10, it is very low, thus not worth considering. Low and medium risks need to be considered. High and very high risks need better treatment as their impacts are high [34,35].

### 4.4. Attack Surface Reduction for Both the Use Cases

A two-step approach is required to reduce the attack surfaces for both the use cases. As a first step, the risks associated with well-known vulnerabilities of the devices are assessed, as shown in Table 8 and Table 9, as well as the well-known threats or attacks associated with both the use cases, as shown in Table 10. Vulnerabilities exist in both the use cases. Having vulnerabilities and threats defined and assessed, it is essential to identify security control methods to patch those vulnerabilities and prevent the attacks. Security controls to reduce the attack surfaces of use cases 1 and 2 are shown in Table 11 and Table 12, respectively.

### 4.5. Attack Surface Reduction in Use Cases 1 and 2

The above table (Table 12) shows the number of vulnerabilities associated with each asset of the use cases that have been exploited by the list of threats to access or damage the assets. This defines the attack surface, the number of all possible points or attack vectors where an unauthorised user can access a system and extract data. Hence, security controls should be applied against the sub-attack surface presented at different layers of the use case to reduce the risk of attack.

### 4.6. Level of Safety for People on Construction Sites

The questionnaire (in Appendix A Table A1) consists of attitudinal questions to be answered with scaling (“strongly agree”, “agree”, “neither agree nor disagree”, “disagree”, and “strongly disagree”) by concerned stakeholders of safety on construction sites including operatives, managers, and safety officers of the construction sites. In the light of data gathered against each question, the analysis will be performed and hopefully provide better safety measures on construction sites [54].

### 4.7. i-TRACE Strength with Blockchain and Digital Twins

The main components of the proposed framework are presented in Figure 6. The participating entities (such as sensors, machines, and humans) registration process is manual as the device IDs are manually added to the code, and any unknown device is ignored (step 1). Next, at the data layer, the assets monitor, collect, and process designated parameters in the physical space of the shopfloor (step 2a). The resource-constrained devices used in the use cases monitor and collect data, whereas the gateways receive requests from sensors requests and process data. The collected data and provenance are sent to the storage layer (step 2b). Provenance is metadata that records a complete lineage of data and a set of actions performed on data [55]. Based on the collected sensor data, domain knowledge, system history data, and process documents, the twinned system generates models and stores them at the storage layer (step 3). The application layer keeps analysing data to detect incidents (step 4). In case of trouble, the respective scheduling services (step 5a) in the physical space or the model calibration services (step 5b) in the virtual space are called, completing the feedback loop. The storage layer provides secure distributed data storage through a lightweight, scalable, and quantum-immune blockchain. In the implementation phase, no actual data values are stored in the blockchain (this is intentional to keep potentially sensitive data private in a potentially public blockchain); instead, only the hash of the data is stored along with information about which data they are so that the device ID, the start time, end time, and the number of expected data points can be verified.

Additionally, surveillance cameras, fire alarms, and power monitoring devices are deployed as physical security countermeasures to prioritise critical events. Since the repercussions of contingency events require immediate actions, we also enable the direct continuous monitoring of such events through sensory data logged at the control unit. Figure 6 further elaborates on the connection between the data layer and storage layer of the proposed framework. Before the production process, necessary details such as sup-ply chain data (e.g., consignment information), order information (e.g., material stock, production quantity, estimated cost), simulation data (equipment historical data, prediction of equipment fault), etc., are already stored in the storage system (step 1a and 1b). Based on the product lifecycle data, domain knowledge, and process documents, the predefined values of the acceptable ranges of system performance parameters are also stored (step 2). We consider steps 1 and 2, a one-time data access during the process.

During the production process, to enforce interoperability, we introduce a data wrangling method which is responsible for cleaning invalid or missing data and converting different data formats into a unified format before inputting data into the twinned system (step 3). As the process initiates, the underlying process data and provenance are recorded on the storage system (step 4). Furthermore, we introduce a data synchronisation method (step 5) for digital–physical mapping and checking for data inconsistencies. The reason for relying on such a process is to limit the frequent, time-consuming access to the blockchain-based storage system, where we explicitly separate the data flow of real-time sensor data and the less dynamic production and provenance data. The data synchronisation method performs a continuous mapping between the predefined equipment performance parameters retrieved from the framework, illustrating the monitoring, collection, storing, processing, and analysis of data from humans, machines, and sensor devices.

Real-time sensor data are obtained from the manufacturing unit to verify their consistency (step 6). The data synchronisation method can access the updated process and provenance them directly from the ledger (other than the manufacturing unit) to eliminate the qualms of those that are untrustworthy. In case of inconsistencies reported by the synchronisation method, corresponding scheduling services (in the physical space) or model calibration services (in the virtual area) carry out the neces-sary measures (step 7a) first at the virtual system (step 7b) and afterwards regulate them on the physical system (step 7c). After resolving the issue, the updated model is stored in the blockchain. The end user or the dashboard uses the blockchain to obtain information about the underlying network, physically and digitally, through edge computing and cloud services.

One clear alternative to integrating blockchain with the IoTs is integrating the IoT and cloud computing [56]. This integration has been used in the last few years to overcome the IoT processing, storage, and access limitations. However, cloud computing usually provides a centralised architecture, which complicates reliable sharing with many participants compared to the blockchain. Integrating blockchain and the IoT addresses previous limitations and maintains reliable data. Fog computing aims to distribute and bring computing closer to end devices, following a distributed approach like blockchain. This can incorporate more powerful machines than the IoT, such as gateways and edge nodes, which can then be reused as blockchain components. Therefore, fog computing could ease the integration of the IoT with blockchain.

### 4.8. i-TRACE Strength Utilizing Blockchain Technology

Since the importance and use of IoTs have been growing exponentially day by day, considering their autonomous and competent intelligent nature in transforming data from the physical to the digital world, the challenges associated with the technology have also been increasing accordingly, particularly in the field of cyber security. The devices in IoT are connected in a decentralised manner, so it would be impractical to use the standard security techniques for communication among the devices. To address the security issues in the decentralised distributed network environment, blockchain technology (BC) is introduced. The BC helps store and provide a decentralised, distributed, and publicly available shared ledger for the processing and verifying of nodes in the network. The data made available in the public ledger are automatically managed by peer-to-peer technology. The BC consists of blocks where the data related to nodes are stored and chained with each other with the help of enclosing the previous block’s hash in the current block, chaining the block together as a circle in the public ledger.

Each block has two sections: the header and the data section, which has a group of transactions. The title presents metadata used to give all the details, such as version number (4 bytes), previous block’s hash (32 bytes), timestamp (4 bytes), Merkle tree (32 bytes), difficulty target (4 bytes), and a nonce (4 bytes) of the block in the ledger. The data section stores and processes data for verification of the transactions. These transactions are the interactions between the nodes in the use cases. The sequence of these transactions presents a use trail of use-device activities in the use cases. The use of BC in the use cases helps to prevent compromise in case of a single point of failure because of its decentralised nature. One of the criticisms raised on using BC technology in resource-constrained devices is that they do not have enough computation or storage power to support them. The answer is that BC considers those devices as a part of their chain that does not have these constraints, such as interaction being carried out through IoT–router in the use cases.

Moreover, traditional IoT environments face some issues, such as scalability, interoperability, security, privacy, trustworthiness, etc. BC has been introduced into the network to address these concerns. The concept of BC comes into existence through the framework of IoT and cloud Integration [57].

### 4.9. i-TRACE Strength Utilising Digital Twins

Digital twins (DTs) were initially presented by the National Aerostatics and Space Administration (NASA) to monitor aerospace missions to diagnose problems and provide proven solutions [58]. Nevertheless, the concept described by the current DTs for simulating real-world systems is not the same as NASA’s suggestion because it is more than just the system’s virtualisation [59]. The concept of DTs, a digital entity of the physical entity, works independently. Still, these two share a twin relationship, being used today, introduced by Michael Grieves in [60] and later in [61]. In connection with these, DTs are considered “machines (physical and virtual) or computer-based models that are simulating, emulating, mirroring or twinning the life of a physical entity” [61]. There are some similar definitions, such as “a system that couples physical entities to virtual counterparts, leveraging the benefits of both the virtual and physical environments to the benefit of the entire system” [61], “multiphysics, multiscale, probabilistic simulation of an as-built vehicle or system that uses the best available physical models, sensor updates, fleet history, etc., to mirror the life of its corresponding flying twin” [62], “a computerised model of a physical device or system that represents all functional features and links with the working elements” [62], and “a virtual representation of real-world entities and processes, synchronised at a specified frequency and fidelity” [59,62].

DTs are integrated multiscale, multiphysics, probabilistic simulations, representations, and real-world mirroring of physical components [62]. DTs are envisioned to transform the working of their products in terms of design and shape and work across various industries. They have a profound impact on different manufacturing industries [59,61]. On the other hand, the IoTs paradigm deploys multiple devices and technologies such as sensors, actuators, microcontrollers, and cloud-enabled services and analytics [63,64]. It is observed that approximately forty-five billion IoT network devices will provide DTs with the data they need in Europe by 2021. According to the report of Gartner [65], 13% of enterprises implemented IoT projects using DTs, whereas 62% are going to employ DTs or planning to do so. They help industries manufacture and manage IoT devices for better and improved outcomes with precise tolerance and flexibility [50]. They also help to comprehend what the IoTs would accomplish before being assembled or manufactured. Because of their striking features, they are expected to be used by NASA and the United States Air Force in future generation vehicles. DTs help perform analytics on holding data, and the need to exchange the enormous amount of data transparently and trustably was a challenging issue [66,67]. Using blockchain with DTs has become the most relevant and capable technology to ensure transparency, trust, and security in DTs [58,68].

Recent advancements in IoT technologies allow DTs to be connected in such a way that supports effective monitoring and data analysis throughout their lifecycle for enabling proactive maintenance, new opportunities development, and planning for further operations with physical counterparts in real-time. The IoT layer stack consists of five layers (Figure 7): physical space, communication layer, digital space, data analysis and visualisation, and application and security layer. The first layer (physical space) consists of sensors, cameras, actuators, etc. The sensors and other intelligent devices collect data from physical objects and send them to the digital space through the communication layer for data storage and processing in the above layers. The second layer (communication) is implemented right above the PSL of the digital twin model. This layer offers to effectively transmit/receive data by the sensors/actuators to the higher layers for further processing and analysis of the data. It acts as a bridge between the physical space and the cyber/virtual space. As it is more likely to consider that both physical exact and virtual space may not be in the same geographical location, the DTs need a wide area wireless network for their communication; in our case, it is considered LoRaWAN. The third layer (virtual or cyberspace) is comprised of two sublayers: data aggregation and modelling. The aggregation layer collects data from underlying sensors through the communication layer for storage and processing. At the same time, the data modelling layer performs modelling of the data present at the aggregation layer. The fourth layer (visualisation) works in connection with the third layer by accessing data from the visualisation layer to mine the data for assessing the condition of the physical objects or systems and predict possible failures and maintenance requirements for the foreseeable future and after performing an analysis on the data reports being made and sent to management. Finally (application and security layers), it is noticed that a large amount of data is being transmitted to the virtual space, which shows that DTs are intrinsically present in a manufacturing process and can dramatically help improve production efficiency, flexibility, and visibility. These node-recorded data can help show the insights of each sensor node and innovative component using business intelligence tools and techniques. For example, the data collected from the physical layer devices can help reduce the devices’ downtimes [69]. Furthermore, the data stored at the visualisation layer are used to present the performance in charts, graphs, and reports. The performance evaluation carried out at the dashboard using such statements helps in identical important key performance indicators (KPIs) to be considered, tracked, and predicted to achieve the goals set by the use cases. Similarly, each layer of DTs can be vulnerable to severe attacks, including demanding, replacing, and stealing sensors, detail of service attacks (injection, sniffing, hijacking, and spoofing), data manipulation, and alteration of analytical algorithms. Cyber-attacks and security threat assessments must be carried out regularly to ensure resiliency.

### 4.10. i-TRACE Strength Utilising the Cloud Environment

The cloud environment provides enormous resources to meet communication and storage requirements through virtualisation. Each request proceeds via an infinite number of processors of the help-constrained devices in IoTs. Cloud computing offers many advantages and services, including Pay-As-You-Go (PAYG), Software-as-a-Service (SaaS), Platform-as-a-Service (PaaS), Infrastructure-as-a-Service (IaaS), Application-as-a-Service (AaaS), and Utility-as-a-Service (UaaS). The merge of cost-effective sensor-based processors with communication technologies brought about the technical revolution in IoTs [70]. IoTs aim to offer direct communication between machines and bring these online to become autonomous, intelligent, and self-organising devices [71].

The IoT devices have been used for monitoring, controlling, or interacting with ubiquitous devices to enable intelligent services such as construction energy management, construction sites, etc. This concept gives birth to the cloud of things (IoT) paradigm [72,73]. Hence, having that paradigm in use provides and manages services and shows great potential to improve the performance and efficiency of service delivery [74]. On the other hand, this architecture tends to be ineffective because of the following challenges. First, the paradigm is based on a centralised communication model, making the network’s scalability harder in case more widespread networks are considered [75]. Second, considering IoT data processing in the paradigm explains that most of the current architecture relies on a third party, hence raising data privacy concerns. The last concern is higher power consumption and communication latency because long data transmission hinders large-scale deployment in practical scenarios. Banafa [76] presented three layers: things, network infrastructure, and cloud infrastructure, as shown in Figure 8 below. Each layer has its issues, as listed in the following diagram.

Figure 1 shows that IoT system architecture comprises the following five components: IoT devices, IoT gateway, router, blockchain, database, and user devices to access the information. In addition to its advantages, in the past decade, the cloud infrastructure has also been singled out for several issues [77]. These issues include security, privacy, losses and risks, scalability, latency, energy consumption, cost, payment, and billing (Table 13) [77].

Given the limitations discussed previously, more sustainable and decentralised applications must be proposed to replace the traditional centralised model with the blockchain, as discussed in the above section. To offer the blockchain in the cloud environment, the computing paradigm helps to achieve the best of both worlds and makes it compatible with sustainable applications for both industries and academia.

Table 13 shows that blockchain offers security, avoids losses and risks, and preserves security and privacy. Blockchain has a robust payment method, while the cloud provides network scalability and combats the forking issues of the blockchain. Issues such as scalability, flexibility, latency, cost concerns, and energy consumption could be addressed by using DTs. It provides a more robust and flexible solution, called tomorrow’s technology, in the form of i-TRACE, certain that the upcoming architecture will be based on some hybrid approach. Based on current ongoing research activities and projects to overcome the challenges, i-TRACE is proposed as a possible hybrid approach and a way forward to overcome existing problems.

### 4.11. i-TRACE Addressed Challenges: Energy Management and Connected Sites

The rapid increase in cyber-attacks is partly due to the phenomenal growth of IoT devices in smart areas such as energy management, construction sites, etc. Security management of the IoT is challenging due to the dynamic and transient nature of the connection between devices [97,100], the diversity of actors capable of interacting within IoT systems [97], and resource constraints [101]. Due to the increasing number of cyber-attacks on IoT devices, maturing security regulations, and rising security concerns, a compound annual growth rate of 33.7% in the cybersecurity market is expected from 2018 to 2023 [95]. Based on recent threat reports, it is further indicated that IoT will be more widespread and impactful and will force senior management to pay more attention to IoT risks while developing the organisation-level cyber risk man-agement [98]. According to the same, only 35% of survey participants reported that they have an IoT security strategy in place, of which only 28% said that they implemented it.

Similarly, another survey [102] presented that 80% of organisations experienced cyber attacks on their IoT devices in the past year. However, 26% of them did not use security protection technologies. These two surveys demonstrate the security limitations many IoT devices have and the need for organisations to proactively invest in IoT cyber security. Despite the weak security measures, existing risk assessment methods are inappropriate for dynamic systems such as the IoTs [103]. i-TRACE used in the energy management and construction site plan scenarios is sufficiently designed for considering interoperability, known vulnerabilities associated with the devices, threat analysis, and cyber security risk assessment of the systems whose complexity broadens wide attack points to adversaries [104]. Developing IoT systems around a standard platform may help organisations develop IoT security measures without inadvertently raising cyber risks [105].

I-TRACE aims to reduce cyber security risks related to energy management and construction site organisations and users by protecting assets and privacy. New technologies are emerging and providing opportunities and challenges for cybersecurity management. However, most of them focus on the technological aspects of IoT cyber security rather than on risk management frameworks to address these complex issues. It has been recognised and endorsed to identify any malicious activity or sources and mitigate attacks in a timely fashion before damaging the organisation’s assets to share cyber threat information (CTI) in a timely and reliable manner. NIST [106] defines CTP as “any information that can help an organisation identify, assess, monitor and respond to cyber threats”. According to a survey conducted by SANS [107], 72% of those responding to the study mentioned that, in 2018, they had produced or consumed such information for their network defence. The respective percentage was 60% in 2017. This shows that information sharing is becoming a part of organisations’ strategies, and the amount of organisations joining the community is rising.

The impact of a successful attack can damage different aspects of the business, which are broadly divided into financial, reputation, and legal aspects. In economic damage, the cyber-attack results in theft of cooperating information, financial information such as bank details or payment card details, etc., theft of money, disruption of trading such as inability to carry out a transaction online, and loss of business or contract. Dealing with these breaches generally incurs costs associated with repairing affected systems and devices. According to the latest government survey [108], 2022 presented that 39% of UK businesses have noticed cyber-attacks. Among these, 31% estimated at least one attack per week, and one in five said that they had experienced adverse outcomes due to an attack; in connection with material products, the average estimated cost of cyber attacks in the last 12 months is GBP 4200. This figure rises to GBP 19,400 for medium and large organisations. At the same time, reputational damage can obliterate the organisation’s trust, an essential element of a customer relationship. This could lead to loss of customers and sales and profit reduction. This can even have an impact on the supplier or affect the relationship ties with the partners, investors, and other invested third parties. Finally, the legal consequences of the cyber breach, as data protection and privacy bind to manage the security of all the personal data being held, may include fines and regulator sanctions if these data are compromised deliberately or accidentally, and the deployment of appropriate security measures failed. Understandably, a security breach can devastate even the most resilient system, but managing the risk of the breach is extremely important in having an effective security incident response in place. It could reduce the impact, report the incident to the concerned authority, clean up the affected area, and have the system up and running in the shortest possible time.

In light of the existing literature on cloud-based IoT, it is accepted that it is more vulnerable to cyber-attacks. To fix those consequences and issues of cloud-based IoT, blockchain-based IoT has been introduced because it has the potential to overcome the problems of cloud-based IoT, including centralisation, by providing peer-to-peer distributed ledger and changes to the stored data, thus providing a high level of trust, immutability, and integrity of data among non-trusting parties. In the case of i-TRACE (a private blockchain), only known participants are admitted to the network and confirm that authenticated and authorised nodes can mine and append the new blocks. Therefore, it ensures that no malicious node exists in the network, increasing the overall security of the system.

To share accountable information among fragmented participants in the digital twin, all transactions must be transparent without any potential manipulation. The shared data needs to be temper-proof and shared as treatable among the participants. However, the issues with DTs concerning data management, including data storage, security, and sharing, still need to be thoroughly realised. Addressing the case above of DTs, using a hybrid approach can selectively store and share important i-TRACE information traceably. The hybrid approach adds authentication and traceability to any transaction that participants share. The hybrid system further uses decentralised mechanisms to authenticate and consent to the accuracy and integrity of transactions among the project participants [109]. This ends up sharing information securely and transparently, making information transmitted in DTs accountable. With the help of this approach, lengthy contracts and payment execution are automated and quickly advanced through the intelligent contract, a self-executing contract protocol intended to facilitate, verify, or enforce an agreement in connection to a contract term automatically. Consequently, reduced collaboration among participants improved project efficiency and customer satisfaction [103]. Moreover, the hybrid approach proved to be the most robust and reliable infrastructure for IoTs [104].

### 4.12. i-TRACE’s Performance Analysis Energy Management and Connected Sites

i-TRACE uses IoT devices that have the feature to communicate through wireless and remote locations and deal with security and privacy challenges, including interoperability, mutability, cyber risk measurement, unrestraint due to no password, devices evolution failure, malicious data access, use of devices in a casual way, etc., in the field of energy management and construction management sites. It supports connected things, humans, systems, and knowledge. It also provides the solution to cyber security disruptions increasing globally due to increasing industrial IoT devices such as sensors and actuators. The devices connected to the internet are being estimated to surpass 20 billion devices [110,111], 30 billion [112], 10 billion [95], and 50 billion [112,113,114,115].

i-TRACE’s security aims to successfully bridge the connectivity, vulnerability, compatibility, interoperability, immutability, attack surface minimisation, and cyber risk minimisation gaps between operational technology (OT) and information technology (IT). The framework for the industry to increase IoT visibility has introduced security issues around link establishment, authentication, key agreement, and encryption methods in IoT applications and services. Security in industrial work was known as safety, and the protection of workers and machines in the industrial setup [22,23] was not associated with the IT world. This means that i-TRACE development of IoT solutions for cyber-physical systems (CPSs) should be the convergence of OT and IT requirements. The IT domain has been identified as a dangerous platform because smart connected devices can compromise industrial systems [24]. The security procedure in the i-TRACE environment should capture the ecosystem’s hardware, networking, application-specific requirements, and third-party aspects.

### 4.13. Comparison of i-TRACE with State-of-the-Art Mechanisms Using Risk Analysis

The i-TRACE project has been observed under different impacts, including safety, health, environment, security, operational disruption, financial/cost of loss, objectives, brand and reputation, legal, regularity, people, technology innovation and delivery, cyber security, likelihood, along with different options, as per Table 6, Table 7, Table 8, Table 9 and Table 10, to calculate the risk against attribute values: very low (1), low (2), moderate (3), high (4), and very high (5), for the particular assets involved for both use cases 1 and 2. It is proved that the i-TRACE solutions engaged for both the use cases and scenarios, with threat agents and their properties presented in Table 14 and Table 15, are more trustworthy and effective for the network’s resilience. It is further observed that i-TRACE is a better solution than existing mechanisms available in the literature, as demonstrated in Table 16 and Table A1 for energy management sites and connected sites, respectively.

The Wilcoxon signed-rank test is used to statistically test the proposed approach and compare it with the state-of-the-art methods for all evaluation metrics. The *p*-value for all evaluation metrics is less than 0.05, indicating that the proposed system is better than the state-of-the-art ones.

## 5. Conclusions

This paper presented the i-TRACE KPIs assessment methodology of IoT devices by employing use cases of energy management and connected sites of CNI. The proposed hybrid methodology is used to combat the growing vicarious nature of remote, well-planned, and well-executed cyber-attacks on CNI’s IoT devices. The i-TRACE KPIs assessment methodology accesses each level of the KPIs of both use cases in detail. The proposed method is divided into four significant steps, including (1) KPI decomposition into different levels, (2) KPIs’ data collection and reporting, (3) KPQs, and (4) KPI assessment performance. These key steps are identified to achieve the desired level of interoperability and immutability of data. A dedicated methodology is also presented herein to assess the KPIs for the i-TRACE project out of the KPAs. The key results areas (KRAs) for both the use cases were assessed and decomposed into smaller, more specific, quantifiable, and measurable indicators as selected SMARTER (specific, measurable, attainable, result-oriented, time-based, evaluate, and re-evaluate) factors. A detailed risk rank calculation method for both use cases, including vulnerability type, impact of CIA traits, exploitability, device risk score, likelihood, risk score, and risk level, was also discussed at the end of the paper.

## Figures and Tables

**Figure 1 sensors-23-08720-f001:**
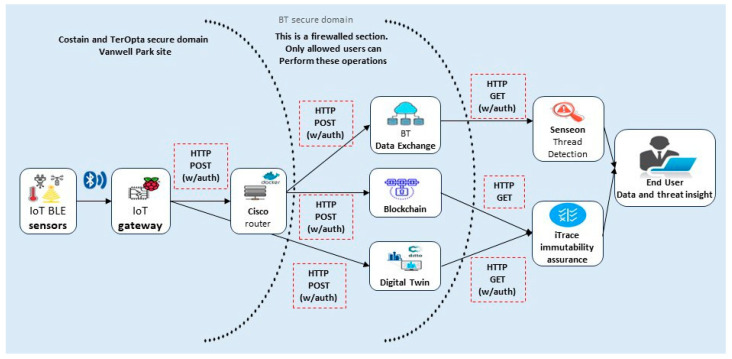
i-TRACE architecture use cases.

**Figure 2 sensors-23-08720-f002:**
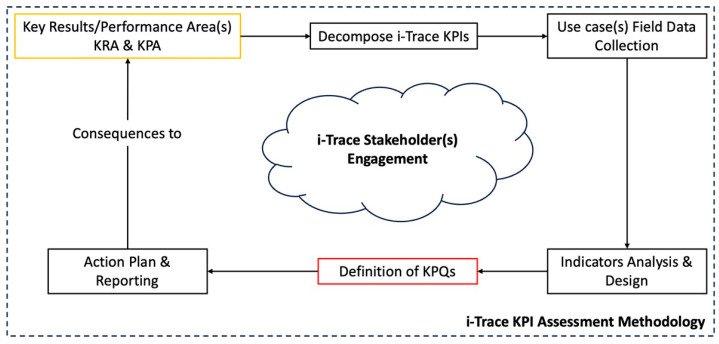
i-TRACE KPIs’ assessment methodology.

**Figure 3 sensors-23-08720-f003:**
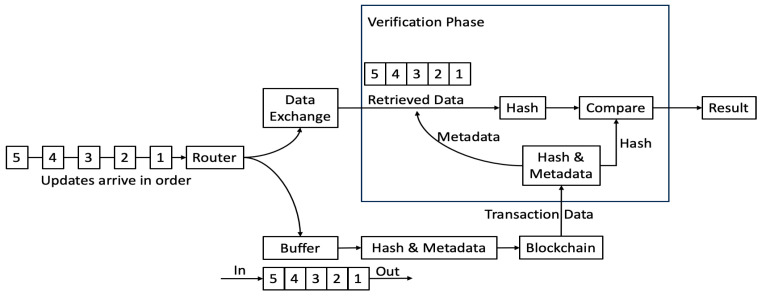
Integration scheme among devices.

**Figure 4 sensors-23-08720-f004:**
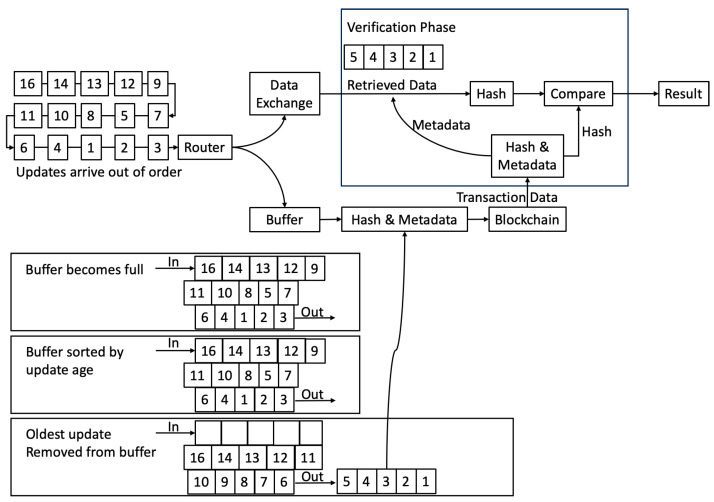
Blockchain communication and calculation of Merkle root.

**Figure 5 sensors-23-08720-f005:**
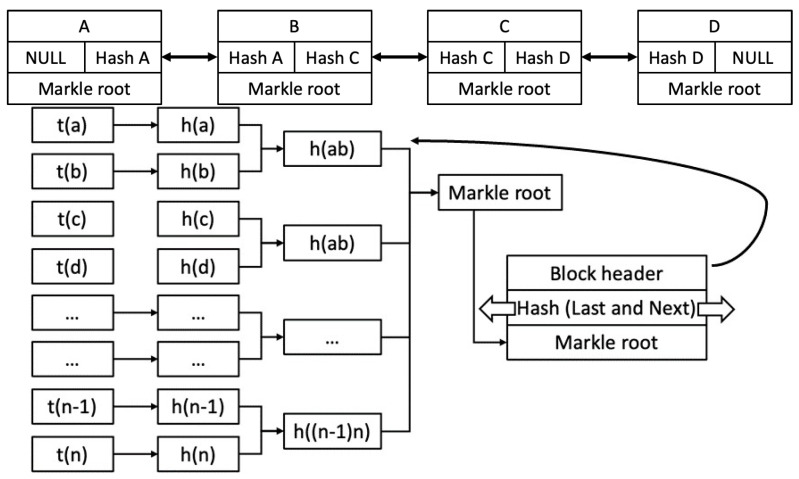
Blockchain communication and calculation of Merkle root.

**Figure 6 sensors-23-08720-f006:**
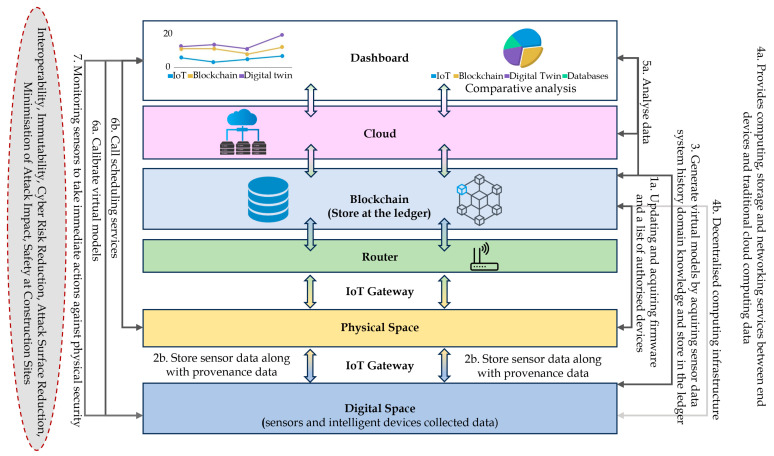
i-TRACE extension(s): cloud and blockchain with DTs.

**Figure 7 sensors-23-08720-f007:**
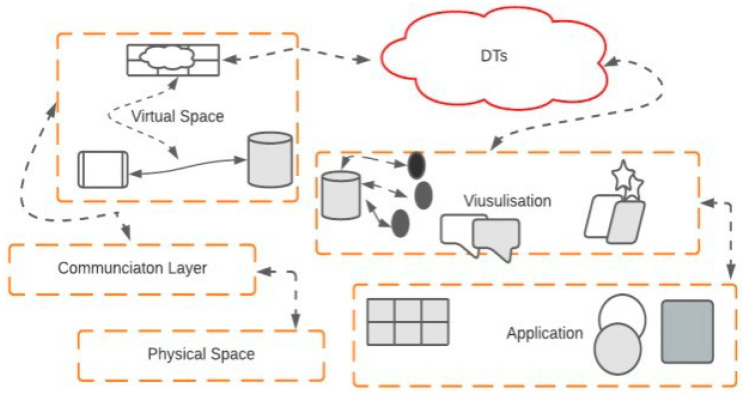
Layers of IoTs: physical to the application layer.

**Figure 8 sensors-23-08720-f008:**
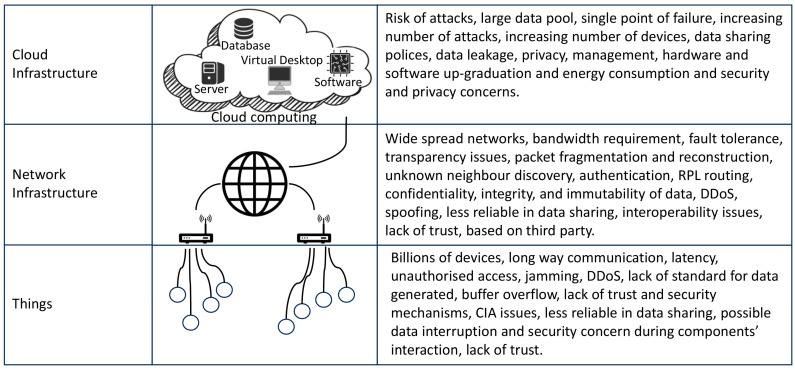
Issues in cloud-computing-based IoT [76].

**Table 1 sensors-23-08720-t001:** i-TRACE and existing architectures: an informal comparison.

Challenges	A. Cloud-Based IoT	B. Blockchain-Based IoT	C. DTs-Based IoT	A + B + C i-TRACE
Security	Low	High	Low	High
Scalability	High	Low	Low	High
Interoperability (within the network)	High	High	High	High
Resilience	Low	High	High	High
Privacy	Low	High	High	High
Data structuring and managing	Low	Low	High	High
Visual representation and simulation	Low	Low	High	High
Losses and risks	High	Low	Low	High
Latency	High	High	Low	Low
Safeguarding product lifecycle	Low	Low	High	High
Cost	Low	High	High	High
Flexible	High	Low	High	High
Decentralised infrastructure	Low	High	Low	High
Immutability	Low	High	Low	High
Transparency	High	High	High	High
Peer-to-peer communication	Low	Low	Low	High
Automation	Low	High	High	High

**Table 2 sensors-23-08720-t002:** KPI assessment criteria.

Criterion	Definition
Quantifiable	The KPI’s value should be numerically specified.
Relevant	The KPI enables performance improvement in the target operation.
Predictive	The KPI can predict no steady-state operations accompanied by a record of past performance values for analysis and feedback control.
Standardised	A standard for the KPI should exist, and that standard is correct, complete, and unambiguous; also, the broader the scope of the standard, the better, for example, plant-wide is good, corporate-wide is better, and industry-wide is best.
Verified	The KPI’s correct implementation can be shown to be true and correct with respect to an accepted standard.
Accurate	The measured value of the KPI is close to the true value.
Timely	The KPI can be computed and is accessible in real-time depending on the operational context. Real-time means the updated KPI is accessible close enough in time to the occurrence of the event triggering a change in any metric affecting the KPI.
Traceable	The steps to fix a problem are known, documented, and accessible, where the particular problem is indicated by values or temporal trends of the KPI.
Actionable	A team responsible for the KPI has the ability and authority to improve the actual value of the KPI within their process.
Buy-in	The team responsible for the target operation can support the use of the KPI and perform the tasks necessary to achieve target values for the KPI.
Understandable	The meaning of the KPI is comprehended by the team members and management, particularly with respect to corporate goals.
Documented	The documented instructions for the implementation of a KPI are up-to-date, correct, and complete, including in-structions on how to compute the KPI, what measurements are necessary for its computation, and what actions to take for different KPI values.
Inexpensive	The cost of measuring, computing, and reporting the KPI is low.

**Table 3 sensors-23-08720-t003:** Key performance indicator analysis for connected and energy management sites for each of the KPIs (#1–6).

KPI #1	
KPI Name	Interoperability SCORE
KPI explanation	The i-Score is an objective function, which we seek to maximise, that represents a summation of spins between all system pairs along the operational thread.
Unit	NUMBER.
Formula	$I=\sum_{i=1}^{n}\sum_{j=1}^{n}mij$
Relevance of the KPI	The goal is to maximise interoperability for an operational thread or set of threads. It is explicitly designed to penalise interoperability function when system pairs need translation in order to interoperate and to reward the interoperability function when their interoperation requires no translation.
Does this KPI affect any part of the scenario?	Because of the heterogeneous environment, increasing interoperability is deemed essential among different devices.
How to measure the KPI?	Using i-Score methodology. The methodology is useful not just to those interested in measuring, analysing, reporting, and improving interoperability of technical systems but is applicable to any situation for which an activity model can be described.
Time to track	On a regular basis.
KPI #2	
KPI Name	Immutability
KPI explanation	Persistence is a basic need of each transaction.
Unit	NUMBER.
Formula	I subsystem 0 mutated and 1 immutable.
Relevance of the KPI	The immutability of data is one of the key properties of blockchain and decentralised authority based on peer-to-peer (P2P) networks. Immutability means that an adversary can no longer hide its tracks or tamper with access logs to erase records of its unwarranted access.
Does this KPI affect any part of the scenario?	Yes, it shows the immutability of data stored on the blockchain.
How to measure the KPI?	If someone tries to alter the data, the system analyses the entire chain and compares, excluding mismatches, thereby preventing unauthorised changes. If changes are made, the immutability value will be 1; otherwise, it will be 0.
Time to track	On a regular basis.
KPI #3	
KPI Name	Reduction in Cyber Security Risks
KPI explanation	The *p* threat is successful based on the level of sophistication and resources. Vulnerability is present and exploitable to produce a material impact. The consequence is the value of the asset(s) at risk.
Unit	Number.
Formula	Cyber risk = threat (intent capability) × vulnerability (target weakness) × (consequence/information value), or R = TVC.
Relevance of the KPI	It is very relevant as cyber security risks are required to be minimised for secure communication between legitimate actors.
Does this KPI affect any part of the scenario?	Yes, it does. The threat, vulnerability, and consequences are required to be calculated and particularly minimised by the use case.
How to measure the KPI?	A successful breach requires an existing vulnerability in the use case that a threat (or bad actor) can find and exploit. An estimate of the value of the underlying asset to be protected is required. What is the cost of the asset’s compromise? When a valuable asset with sensitive data or a client that has access to those data has a vulnerability that can be exploited, the consequences can be significant.
What is the cost of that asset becoming compromised?	When a valuable asset with sensitive data or a client that has access to those data has a vulnerability that can be exploited, the consequences can be significant.
Time to track	Quarterly.
KPI #4	
KPI Name	Level of Safety for People on Construction Sites
KPI explanation	We can calculate it by reviewing the literature published on safety, followed by exploratory interviews, which take place with two operatives, two site managers, and one safety officer on site. The interview discussions will be focused on the causes of accidents and the attitude of workers toward safety on site. After the exploratory interviews, a pilot study questionnaire will be designed. Each questionnaire consists of 34 questions which relate to the research variables, namely, historical information (V1), economical (V2), psychological (V3), technical (V4), procedural (V5), organisational (V6), and environmental (V7). Safety performance (V8) is identified as an accident occurrence to a person resulting in various degrees of injury.
Unit	Number.
Formula	r=∑(xi−x^)(yi−y^)∑(xi−x^)2∑(yi−y^)2 Here, *n* = the number of pairs of scores, Σ*xy* = the sum of the products of paired scores, Σ*x* = the sum of x scores, Σ*y* = the sum of y scores, Σ*x*^2^ = the sum of squared x scores, and Σ*y*^2^ = the sum of squared y scores.
Relevance of the KPI	It helps to have the safety of people in the construction site in place, to mitigate the most important concerns.
Does this KPI affect any part of the scenario?	Yes, it does, especially for the construction sites.
How to measure the KPI?	SPSS (Statistical Package for Social Science). Two statistical techniques were used: the Pearson’s correlation coefficient (for linearity) and the factor analysis (for nonlinear groupings). Pearson’s correlation measures the strength of the relationship between the research variables and safety performance.
Time to track	Quarterly.
KPI #5	
KPI Name	Reduction in Attack Surface (Attack Surface Analysis)
KPI explanation	The attack surface includes all the cases in which an attacker could compromise the devices used in the use case or networks. Reducing attack surface means protecting the use case’s devices and network, which leaves attackers with fewer ways to perform attacks.
Unit	%
Formula	Σ surface area (SA) score% = Σ SA (baseline proposed/idea performance) /Σ SA (actual achieved/real performance). The effectiveness scores of before and after the improvements are compared by taking the average of the total scores. Reduction in SA = Σ average {pre-improvisation effectiveness} − Σ average {post improvisations effectiveness}.
Relevance of the KPI	This is used to help in measuring the surface area (SA) and then reducing it accordingly.
Does this KPI affect any part of the scenario?	Reducing the attack surface means protecting the deployed devices and network in the use case, which leaves attackers with fewer ways to perform attacks.
How to measure the KPI?	Reducing the threat surface area by measuring the security vulnerabilities to produce a score first and then reducing the service benefits obtained when exploiting the resource.
Time to track	Quarterly.
KPI #6	
**KPI Name**	Level of Minimisation of Attack Impacts
KPI explanation	Data collected from various available resources at the site, and analysis will be made from those collected data. It will be required to be understood based on the collected data to decide which type of cyber-attacks occurred. According to the general investigation, it has been examined that more than 50% of the energy management site was apparently affected by the following major five cyber threats: denial of service (DOS), phishing, malware, spear phishing, and ransomware.
Unit	%
Formula	Per = (individual cyber-attack type/collected data) × 100.
Relevance of the KPI	It helps to minimise attack impacts by considering the importance and severity of the data.
Does this KPI affect any part of the scenario?	It measures and improves the overall system site by tracking incidents that must be handled on a priority basis.
How to measure the KPI?	SPSS (Statistical Package for Social Science). Two statistical techniques were used, namely, Pearson’s correlation coefficient (for linearity) and the factor analysis (for nonlinear groupings). The Pearson’s correlation measured the strength of the relationship between the research variables and safety performance.
Time to track	Quarterly.
Traceable	Results will be stored for future use.

**Table 4 sensors-23-08720-t004:** Table representing values for interoperate score.

Device (i)	Device (j)	(IJ)	Interoperate Scope (s)
Sensor	IoT gateway	(1, 2)	*sij* = 1
IoT gateway	Cisco router	(2, 3)	*sij* = 1
IoT gateway	Digital twin	(2, 4)	*sij* = 1
Cisco router	Database	(3, 5)	*sij* = 1
Cisco router	Blockchain	(3, 6)	*sij* = 1
Database	Sensor Al/ML	(5, 7)	*sij* = 1
Database	Dashboard visualisation	(5, 8)	*sij* = 1
Dashboard	Visualisation sensor	(8, 1)	*sij* = 1

**Table 5 sensors-23-08720-t005:** Communication at different levels and its impact.

Criteria	(a). Decentralised Distributed Blockchain Gateway Level	(b). Decentralised Distributed Blockchain IoT Level	(c). Distributed Blockchain Edge Devices Level	(d). Cloud Blockchain Hybrid with the IoT Edge
Requirements	Gateways should be registered	IoTs need to register	Should all part register on network	No need for all parties to register, just those that are in blockchain
Immutability	Very High	High	High	High
Security	Very High	High	High	High
Traceability	Low	Low	High	High
Privacy	Low	Medium	Medium	Medium
(a). The IoT devices, such as BLE sensors, are registered to the gateway device, which performs transactions to the blockchain on behalf of these devices. This approach enables the traceability of all communications involving a specific IoT gateway and services. This integration scheme can also be used to authenticate communications between devices connected to separate blockchain-enabled gateways				
(b). Interconnected edge devices as end-points to the blockchain are similar to the previous approach, and all IoT interaction events are logged into the blockchain for secure accountability. Cryptographic functionality can be also provided. The trade-off here is a higher degree of autonomy of IoT devices and applications versus increased computational complexity of the IoT hardware.				
(c). IoT gateways and devices issue transactions to the blockchain and can communicate with each other. This approach ensures low latency between the IoT devices and chooses specific interactions on the blockchain				
(d). his method leverages the benefits of decentralised record keeping through blockchains as well as real-time IoT devices’ communication. Due to this hybrid integration schema, the challenge posed by this approach is to optimise the split between the interactions that occur in real-time and those that go through the blockchain. Also, it can utilise cloud computing to overcome the limitations of blockchain-based IoT networks, such as storage [20].				

**Table 6 sensors-23-08720-t006:** The devices used in the use cases energy management and connected sites.

Use Case 1: Devices used in Connected Sites
3× 4 K 40× Starlight IR PTZ AI Network Camera (DH-SD8A840WA-HNF), 3× Outdoor temperature/humidity, 3× Vibration monitor, 5× Outdoor GPS sensors, 1× Light level sensor, 1× Door/window open, 1× Wind sensor, 3× Noise sensor, Milesight UG67 Gateway (LoRaWAN Network Server), Libelium- Plug and Sense! SE-PRO LoRa, Libelium—Carbon Monoxide (CO) Low Concentration, Libelium—Nitro Dioxide (NO_2_) [Calibrated] (High Accuracy) Probe, Libelium—Ozone (O_3_) [Calibrated Probe 9374-P, Libelium-Sulphur Dioxide (SO_2_) [Calibrated] (High Accuracy Probe 9377-HA-P, Libelium—Temp, Humidity and Pressure Robe 9370-P, Libelium—Particle Matter (PM1/PM2.5/PM10)-Dush Probe 9387-P, Libelium—External Solar Panel 7v-500 mA (Power Accessory for P&S) PAPS-ESP, Libelium-220 V adapter+ outdoor USB cable (power accessory for P&S!) PAPs-220 V-OUT, Ranos dB2 Sound Sensor RANOS DB-2, BOB Assistant 6-Axis accelerometer and gyroscope BOB-EU868, Abeeway Compact Tracker ABEEWAY-COMPACT-TRACKER-EU868, Milesight- UC501-868M IO Controller UC501-868M, (MTGMISC) Shipping Charges SHIPPING, i-TRACE, 1× Cisco IR1101 Industrial Integrated Services Router Rugged, 1× Cisco IE 1000-4P2S-LM Industrial Ethernet Switch.
**Use Case 2: Energy Management System**
i-REAP, 12× Gateways based on Raspberry Pi Model 3B+, 74× BLE Sensors (5), 2× TerOpta GEM energy monitoring and building control units, 1× Outdoor wired temperature and relative humidity sensor (RH632), 1× Solar irradiance sensor (SI–V–10TC), 1× Wind-speed sensor, i-TRACE 1× Cisco IR1101 Industrial Integrated Services Router Rugged.

**Table 7 sensors-23-08720-t007:** Level of risk rank criterion. The different colours represent the different levels of IoT risks, with black being the high probability of risks and blue for no risk concern.

Qualitative Level	Quantitative Weightage (W)	Risk Score (S)	Rank = W × S	Risk Rank Range	Presentation Colour	Description
Very high	96–100	1.0	97 × 1.0 = 97	81–100		Risk is of very high concern; severe impact.
High	80–95	0.8	90 × 0.8 = 72.	51–80		Risk is of high concern.
Medium	31–79	0.5	50 × 0.5 = 25	21–50		Risk is of moderate concern.
Low	11–30	0.2.	25 × 0.2 = 5	5–20		Risk is of low concern.
Very low	0–10	0.1	10 × 0.1 = 1	0–4		Risk is not of concern.

**Table 8 sensors-23-08720-t008:** Use Case 2: Energy management’s risk rank calculation.

Device	Vulnerability Type	Impact (CIA) (Imp)	Exploitability (Exp)	Device Risk Score (drf)	Likelihood (Lik) Exp + drf/2	Risk Score Imp × Lik	Risk Level
Gateways based on Raspberry Pi Model 3B+	Does not correctly verify the ownership of a communication channel.	6.4	10	7.5	8.75	56	High
	Denial of service overflow.	10	3.9	7.2	5.55	56	High
	Allow an unauthenticated, remote attacker.	2.9	8.6	4.3	6.45	19	Low
	Execution of code.	10	10	10	10	100	Very High
	Spamming attack.	2.9	10	5	7.5	22	Medium
	Attacker to cause a denial of service (DoS) condition on an affected device.	10	3.9	7.2	5.55	56	High
	Physical access security vulnerability.	10	3.9	7.2	9.15	92	Very High
BLE Sensors	Enables an attacker with user-level access to the CLI.	10	8	9	8.5	85	Very High
	Allows remote attackers to execute arbitrary code.	6.4	6.5	5.8	6.15	39	Medium
TerOpta GEM energy monitoring and building control units	Allow an unauthenticated access.	2.9	10	5	7.5	22	Medium
	Vulnerable to remote code execution.	10	10	10	10	100	Very High
	Allows an attacker to perform an MITM.	5.9	1.6	7.5	4.55	29	Medium
	An attacker can use this overflow to gain full control.	10	5.5	7.9	6.7	67	High
Wind sensor	DoS.	2.9	10	5.0	7.5	22	Medium
Solar irradiance sensor (SI–V–10TC)	Remote attackers can gain privileges and execute arbitrary code.	10	10	10	10	100	Very High
Outdoor wired temperature and relative humidity sensor (RH632))	Gives the attacker control over the data that are written into this doubly allocated memory.	5.9	1.8	7.8	4.8	28	Medium
Cisco IR1101 Industrial Integrated Services Router Rugged	Allow an authenticated but low-privileged local attacker.	10	3.9	7.2	7.5	75	High
	An unauthenticated attacker with physical access to the device opens a debugging console.	5.2	0.9	6.1	3.5	18	Low
	Allowing an attacker with administrator privileges to access sensitive login credentials or reconfigure the passwords.	5.2	0.3	5.5	5.65	29	Medium

**Table 9 sensors-23-08720-t009:** Use Case 1: Connected site’s risk rank calculation.

Device	Vulnerability Type	Impact (CIA) (imp)	Exploitability (exp)	Device Risk Score (drf)	Likelihood (Lik) Exp + drf/2	Risk Score Imp × Lik	Risk Level
IR PTZ AI network camera	Improper access control.	2.9	10	5	12.57	36	Medium
	Denial of service and execute code overflow.	10	8.6	9.3	8.95	90	Very High
Outdoor temperature/humidity	Denial of service.	7.8	10	8.5	9.25	93	Very High
Outdoor GPS sensors	Weak encryption scheme.	2.9	10	5.0	7.5	22	Medium
Light level sensor and Door/window open	Intentional or unintentional disclosure of information.	2.9	10	5.0	7.5	22	Medium
Noise sensor	Cross-site scripting	2.9	6.8	3.5	8.5	25	Medium
Milesight UG67 Gateway LoRaWAN (Built-in LoRaWAN network server)	Do not validate or incorrectly validate input that can affect the control flow or data flow.	6.9	8.0	6.8	7.4	51	High
	Denial of service.	2.9	10	5.0	7.5	22	Medium
Libelium Environment Pro LoRaWAN EU 868	Unintentional disclosure of information to an actor that is not explicitly authorised to have access.	2.9	3.9	2.1	3	09	Low
	Execute code memory corruption.	2.9	8	4.0	6	17	Low
Libelium Carbon Monoxide Low Concentration 9371-lc-p	Do not validate or incorrectly validate input that can affect the control flow or data flow.	4.9	8.0	5.5	6.75	33	Medium
Libelium Nitric Dioxide (NO_2_) [Calibrated] (High Accuracy) Probe 9376-HA-P	Do not properly restrict the size or number of resources that are requested or influenced by an actor.	6.9	10	7.8	8.9	61	High
Libelium Ozone (O_3_) [Calibrated] (High Accuracy) Probe 9377-HA-P	Denial of service (DoS).	6.9	10	7.8	8.9	61	High
Libelium Sulphur Dioxide (SO_2_) [Calibrated] High Accuracy Probe	Compromises CIA of the device.	10	3.9	7.2	5.55	56	High
Libelium Temp, Humidity and Pressure Probe	Denial of service or information disclosure.	4.9	10	6.4	8.2	40	Medium
Libelium External Solar Panel 7 V-500 mA (Power Accessory for P&S)	Denial of service (DoS) attack.	10	3.4	6.9	5	52	High
Ranos-dB2-Sound sensor	Allows an attacker to decrypt highly sensitive information.	2.9	10	5.0	7.5	22	Medium
BOB Assistant 6-Axis accelerometer and gyroscope	Could allow an authenticated, remote attacker to conduct SQL injection attacks.	4.9	8.0	5.5	6.75	33	Medium
Abeeway Compact Tracker-LoRaWAN GPS Tracker	Information leak and denial of service conditions.	4.9	10	6.4	11.4	56	High
Milesight-UC501-868 IO Controller	Unauthenticated attacker to trigger a denial of service.	2.9	10	5.0	7.5	22	Medium
(MTGMISC) Ship- ping charges	Unauthorised commands.	10	10	10	10	100	High
Cisco IR1101 Industrial Integrated Services Router Rugged	Allows attackers to execute unexpected, dangerous commands.	10	3.9	7.2	5.55	56	High
Cisco IE 1000-4P2S- LM Industrial Ethernet Switch	Allows an unauthenticated, remote attacker to conduct a cross-site request forgery (CSRF) attack.	6.4	8.6	6.8	7.7	49	Medium

**Table 10 sensors-23-08720-t010:** Known vulnerability associated with both the use cases.

Vulnerability No	Description	Common Vulnerability Exposure (CVE)
V1	Poor physical security.	(CVE-2020-7207, CVE-20151231, CVE-2014-9689, CVE-2019-18618)
V2	No energy harvesting.	(CVE-2020-13594)
V3	Open debugging ports.	(CVE-2019-10939)
V4	Poor/hand-coded password.	(CVE-2021-22729), (CVE2021-27254) (CVE-2021-32525)
V5	Boot process vulnerabilities.	(CVE-2021-1398), (CVE-20201073)
V6	Total loss of availability.	(CVE-2021-34398), (CVE2020-12826)
V7	Improper encryption.	(CVE-2022-24318), (CVE-2021-38983)
V8	Improper patch management.	(CVE-2021-44228)
V9	Insecure network services.	(CVE-2020-10281), (CVE-2007-3026)
V10	Insecure ecosystem interfaces.	(CVE-2022-22331), (CVE-2017-7577)
V11	Weak authentication/authorisation.	(CVE-2000-1179), (CVE-19991077), (CVE-2002-0066)
V12	Insecure cloud interface.	(CVE-2021-22914), (CVE-2022-23105)
V13	Missing authorisation.	(CVE-2022-24317), (CVE2019-3399), (CVE-2022-26102)
V14	Unencrypted services.	(CVE-2020-25178), (CVE2019-18285), (CVE-2020-25178)
V15	MiTM attackers.	(CVE-2015-4000)
V16	Allowing attackers to execute arbitrary commands.	(CVE-2017-5638)
V17	Giving read–write access with root privileges.	(CVE-2018-15664)
V18	Overwriting the host runc binary.	(CVE-2019-5736)
V19	Docker allows remote authenticated users to cause a DoS.	(CVE-2016-6595)
V20	Unauthenticated remote attacker.	(CVE-2018-7445)
V21	Allows an attacker to intercept the database connection or have read access to the database.	(CVE-2020-5899)
V22	Attacker affects all hardware wallets.	(CVE-2020-14199)
V23	Integer overflow of a smart contract.	(CVE-2021-34270)
V24	Attackers to prevent authorised users from monitoring the BGP status.	(CVE-2020-3449)
V25	Authenticated and unauthenticated pairing with both LE secure connections.	(CVE-2020-11957)
V26	Allowing untrusted applications to access the Bluetooth information in a Bluetooth application.	(CVE-2021-25430)
V27	Allowing attackers in radio range to cause an event deadlock or crash.	(CVE-2019-19192)
V28	Allowing to masquerade as another user.	(CVE-2020-12691)
V29	An attacker can steal a user’s session ID to masquerade as a victim user.	(CVE-2021-25926), (CVE-2021-25926)
V30	Zero-day vulnerability.	(CVE-2022-26143)
V31	DOS attacks to congest traffic or drain sensor battery.	(CVE-2017-7670), (CVE-2022-26143)
V32	Inject false data as part of an attack further down the chain.	(CVE-2017-5638), (CVE-2021-45901)
V33	Improper access control vulnerability.	(CVE-2022-23433)
V34	Eternal Blue Exploit:	(CVE-2017-0144)
V35	BlueBorne (security vulnerability).	(CVE-2017-14315)
V36	Information leaking.	(CVE-2017-0785)
V37	Bluetooth hack.	(CVE-2018-5383)
V38	Bluetooth data leak (protocol).	(CVE-2020-29531)
V39	Predictable AuthValue in Bluetooth Mesh Profile provisioning leads to MitM.	(CVE-2020-26557)
V40	Impersonation in the Passkey entry protocol.	(CVE-2020-26558)
V41	Bluetooth Mesh Profile AuthValue leak.	(CVE-2020-26559)
V42	Impersonation attack in Bluetooth Mesh Profile provisioning.	(CVE-2020-26560)
V43	Impersonation in the BR/EDR pin-pairing protocol.	(CVE-2020-26555)
V44	Malleable commitment in Bluetooth Mesh Profile provisioning.	(CVE-2020-26556)
V45	Impersonation in the BR/EDR pin-pairing protocol.	(CVE-2020-26555)
V46	Attackers intercept and manipulate Bluetooth communications/traffic between two vulnerable devices.	(CVE-2019-9506)
V47	Affects the Bluetooth BR/EDR (basic rate/enhanced data rate) key negotiation procedure/protocol.	(CVE-2019-9506)
V48	An attacker can use this overflow to gain full control of the device through the relatively high privileges of the Bluetooth stack.	(CVE-2017-14315)
V49	An attacker might be able to allocate the overwritten address as a receive buffer, resulting in a write-what-where condition.	(CVE-2019-13916)
V50	An attacker can steal a user’s session ID to masquerade as a victim user.	(CVE-2021-38759)
V51	Gateway does not correctly verify the ownership of a communication channel.	(CVE-2019-9010)
V52	A network port intended only for device-internal usage is accidentally accessible via external network interfaces.	(CVE-2021-20999)
V53	Remote attackers are allowed to gather information about the file system structure.	(CVE-2019-11602)
V54	Remote attackers are allowed to read files outside the http root.	(CVE-2019-11603)
V55	The IoT Message Gateway Server is affected by a buffer overflow vulnerability.	(CVE-2020-4207)
V55	Hackers are allowed to look for and eventually gain access to sensitive files.	(CVE-2017-7577)
V56	There are hard-coded system passwords that provide shell access.	(CVE-2021-33218)
V57	Attackers are allowed to impersonate.	(CVE-2021-28372), (CVE-2018-8479)
V58	Remote attackers are allowed to bypass access controls.	(CVE-2009-0801), (CVE-2021-34739)
V59	In an unauthenticated situation, remote attackers are allowed to retrieve sensitive information.	(CVE-2019-1653)
V60	An attacker who can control log messages or log message parameters can execute arbitrary code.	(CVE-2021-44228)

**Table 11 sensors-23-08720-t011:** Security control methods to address the vulnerabilities.

Security Control	Type	Description
Sc1	Physical security	Physical security devices such as door locks, security guards, access control cards, and fire suppression systems protect the physical location. These locations should also be monitored by devices such as surveillance cameras, smoke detectors, heat detectors, and intrusion detection sensors. This ensures that IoT devices will not be destroyed or, worse yet, stolen [34,35]. If an attacker were to gain physical access to the device, they would gain the ability to do anything that they want with it regardless of any other countermeasures that might be put into place. It gives them limitless time to break passwords or try different methods to gain access to the information that they would not have been able to do otherwise due to time constraints created by the risk of being caught during their attacks.
Sc2	Authentication of devices	Authentication of the devices is required to be ensured before getting into the use case in order to keep the malicious devices from accessing part of the network so that forged data following in the network could be prevented [36].
Sc3	Secure physical designing of end devices	Perception, sensor layers, or attacks can be resolved by the secure physical design of end devices. The components of devices, such as radio frequency circuits, chip selection, etc., must be of high quality. For example, an antenna with a good wireless communication design could be able to communicate over a long distance [37].
Sc4	Safe booting	A cryptographic hash algorithm can be utilised to check the integrity and the authentication of the software on different devices of the use cases. In fact, most of the hash algorithms cannot be implemented on network end devices because these devices possess very low computing power; therefore, WH and NH cryptographic algorithms are the optimum solutions to this problem [37].
Sc5	The integrity of data	Each device utilised in the environment should be provided with error detection systems such as a checksum, a parity bit, etc., to decrease the risk of data tempering; the cryptographic hash function should be used to make the network more secure [37].
Sc6	Anonymity	An attacker can hide classified information, such as identity, location, etc., by injecting a node into the network. The K-anonymity approach is the best solution to this problem [38] as it works better on low-processing devices [35].
Sc7	DoS protection	A denial of service (DoS) attack occurs when an attacker attempts to overwhelm a target machine (e.g., a server) by sending a stream of data packets so that authentic users cannot access it [39]. DoS protection can detect the attack and prevent overwhelming the system. Protection can come in the form of a physical appliance or configured software (e.g., a firewall). It can also be offered as a service from a provider filtering traffic that follows certain patterns. It is important for the companies to be aware of the usual amount of traffic their sites receive at various times. Thus, whenever there is a massive spike in traffic, to detect it early and mitigate some of the damage. Several methods to exist to prevent DoS attacks that work by monitoring incoming traffic. These can include filtering traffic from a specific IP address and limiting how many packets can be sent from an individual IP address, as well as forwarding any packets from specific IP addresses and dumping them without allowing them to reach their intended target [40].
Sc8	Event reporting	Event reporting keeps a log of all abnormal or unusual activities, such as login attempts, on devices. Specifically, it refers to reporting suspicious or anomalous activities, ranging from an incorrect password to an attempted breach to noncompliance. Even if it ends up being an innocent mistake, taking this pre-emptive step can prevent future disasters. Event logging and reporting allow companies to notice when anything out of the ordinary occurs so they can take steps to address it.
Sc9	Data encryption	As the name implies, is used to encrypt the data to be protected even if an unauthorised user intercepts the encrypted data packets over a communication channel. Current enterprise-grade encryption standards can take years of computing power to crack. The best way to avoid a man-in-the-middle attack is to use a robust encryption method from the client and the server. It is also important to implement some form of nonrepudiation [29,30,31,32,33,34,35,36,37,38,39,40,41,42]. IoT manufacturers should focus on identity and authentication when producing devices and sending them to the market. Another method is by using an encrypted secured channel with the use of a Virtual Private Network (VPN) as a communication tunnel between two or more devices and encrypting anything in and out of the tunnel. With this solution the attacker will not be able to read the data when they monitor the communications [43].
Sc10	Offsite data backups	Conventional data backups prevent the loss of data as they are stored off-site or are air-gapped. That approach can be vital in preventing loss of operational capacity in a catastrophic incident. For example, if the entire organisation is compromised by ransomware, restoration from a backup can fix the immediate problem. This is essential for companies that keep records of various transactions, such as banks but also prevents data loss from any natural disasters such as floods or earthquakes.
Sc11	Input sanitisation	It prevents code from being put into input fields, which could have a hazardous effect on databases connected to the system [44]. It is very easy to neglect, and many major companies have succumbed to injection attacks due to a missing code line. Anywhere a user can type in information needs to be sanitised appropriately, regardless of how innocuous it may seem. Vulnerable areas can range from a search bar to a login field to a page for people to leave comments. This prevents attacks such as SQL injection from occurring on any systems that a user has access to [45].
Sc12	Intrusion detection/prevention	Intrusion is the attempt to gain access to unauthorised systems or resources [46]. Intrusion detection/prevention detects/prevents unauthorised users from accessing various parts of the network. It can also alert the company about the intrusion [47]. The difference between intrusion detection and prevention is obvious by the used term. Intrusion detection monitors traffic or observes system behaviour for anything that might be malicious (e.g., policy violations and anomalies) but does not necessarily take immediate action to stop the potential intrusion. Intrusion prevention, on the other hand, actively prevents and stops any intrusions from reaching the target server or any valuable resource of the private data network.
Sc13	Confidentiality of data	Data confidentiality can be ensured by preventing illegitimate access to the nodes of the IoT network. Point-to-point encryption can be utilised for authentication purposes. In this process, classified data are immediately converted into cypher code, which is unbreakable.
Sc14	The integrity of data	Applying a cryptographic hash function on the data verifies that it is not tempered on reaching the receiving side, thus ensures data integrity.
Sc15	Secure routing	Secure routing plays a vital role in the safe usage of sensor systems as most of the routing conventions are not stable. Routing the data through several paths increases the network’s error exposure.
Sc16	Spoofing	GPS location systems can face spoofing attacks. For this problem, a perfect solution has yet to be provided; however, S. Daneshmand et al. [47] described the GPS system techniques, which are the best.
Sc17	Inside and outside attacks	Attacks from inside the network can be secured by security-conscious ad hoc routing modus operandi. Attacks from outside the network can be secured by encryption and authentication so that the hacker cannot join the network.
Sc18	Encryption to secure classified information	The primary purpose of encryption is to secure sensitive data by storing or transmitting them to the cloud in encrypted form to prevent security breaches. Today, various types of encryption methods are used, which help defeat side-channel attacks and secure the use case.
Sc18	User validation	Integrity and encryption mechanisms are vital for the security and privacy of a system because data stealing and unauthorised access to the use case can cause a security breach.
Sc19	Bluetooth security practices	Default settings should be updated to achieve optimal standards [48]. Ensuring devices are and remain in a secure range by setting the devices to the lowest power level [49]. Using long and random PIN codes makes the codes less susceptible to brute-force attacks [49]. Change of the default PIN and frequently updates of this PIN (i.e., once every other month). Setting devices to undiscoverable mode by default, except as needed for pairing [49]. Most active discovery tools require devices to be in a discoverable mode to be identified. Devices set to undiscoverable mode will not be visible to other Bluetooth devices. However, they will still be able to connect and communicate with devices previously configured, known as trusted devices. Pairing devices as needed [43]. Any pairing should take place in a secure, non-public setting [49]. This will prevent attackers from intercepting pairing messages [48]. Pairing is a crucial part of Bluetooth security an d users should be aware of eavesdropping [49]. When possible, SSP should be used instead of legacy PIN authentication for the pairing exchange process. This will mitigate PIN cracking attacks. Turning off a device’s Bluetooth when not needed or in use, especially in certain public areas such as shopping malls, coffee shops, public transportation, clubs, bars, etc. [5]. This can prevent users from receiving advertisements from other Bluejackets. Refraining from entering passkeys or PINs when unexpectedly prompted to do so. Frequent software updates and recent drivers to have the most recent product improvements and security fixes. Refraining from using non-supported or insecure Bluetooth-enabled devices or modules, including Bluetooth versions 1.0 and 1.2. All lost or stolen Bluetooth devices should be unpaired from previously paired devices [50]. Unpairing will prevent an attacker from accessing the users’ other devices through Bluetooth pairing [49]. Users should never accept transmissions from unknown or suspicious devices [49]. Content should only be accepted from trusted devices [49]. All paired devices should be removed immediately after use. Devices should be monitored and kept at close range.
Sc20	Man-in-the middle	Combination keys should be used instead of basing link keys on unit keys to prevent man-in-the-middle attacks [48,49].
Sc21	Eavesdropping	Link encryption should be used for all data transmissions to prevent any eavesdropping, including passive eavesdropping [43]. Using the HID boot mode mechanism, a connectionless human interface device should be avoided, as it sends traffic in plaintext.
Sc22	Enabling encryption	Users should ensure all links are encryption-enabled when using multi-hop communication [49]. Failure to do so could compromise the entire communication chain [49].
S23	Mutual authentication	Mutual authentication is required for network-connected devices [8]. This will confirm that the network connections are legitimate [49].
Sc24	Broadcast interceptions	The risk of broadcast interceptions decreases by encrypting the broadcasts [49].
Sc25	Maximum encryption key size	The maximum encryption key size should be used [3]. In addition, a minimum key size should also be set—128 bits as it was recommended in [49]. The utilisation of these minimum and maximum keys will protect devices from brute-force attacks [49].
Sc26	Bluetooth security	Security Mode 3 is highly recommended to provide the highest level of security [49]. This mode of security, implemented at the link level, is one of the highest levels of Bluetooth security [49].
Sc27	Gattacker	Multiple MITM detection and mitigation metrics were proposed. A proof of concept is provided in the research presented in the article [48].
Sc28	Infect device over Bluetooth BlueBorne poisoning, protocol fuzzing	The proposed method can complement the security method for systems and services based on BLE [51].
Sc29	LoRaWAN channel confidentiality	A solution is suggested in [52].
Sc30	Spoofing LoRaWAN	A solution is suggested in [52].
SC31	Attack to the VPN	A solution is suggested in [53,54].

**Table 12 sensors-23-08720-t012:** Security controls are defined for the use cases 1 and 2.

Vulnerability Exploited	Threats/Attacks Use Case 1	Security Controls	IoT Layer	Use Case Devices
V1, V2, V3, V4, V5, V6, V7, V9, V10	Hardware attacks by changing power control, modifying settings of devices to disrupt the system, Mascaraed, DOS attack to congest traffic or drain sensor battery, injecting false data as part of an attack further down the chain, incorrect control commands to damage sensors, blue sniffing, stealing information over the Bluetooth protocol, Eavesdropping, Gattacker BLE MiTM enabling data modification, infecting a device over Bluetooth BlueBorne poisoning, protocol fuzzing	Sc1, Sc2, Sc3, Sc4, Sc5, Sc6, Sc7, Sc8, Sc9, Sc10,	Physical layer	Sensors
V4, V6, V7, V8, V9, V10, V11, V12, V13, V14	Impersonation (fake input data representing real sensor), spoofing (LoRaWAN uses default A15-128 keys, tampering (integrity) DoS (interferences), physical access (authentication is a single factor), attack to the VPN, spoofing DoS, bug exploitation in the SC code, elevation of privileges.	Sc2, Sc4, Sc5, Sc6, Sc7, Sc11, Sc14, Sc18, Sc19, Sc20, Sc21, Sc22, Sc23, Sc24, Sc25, Sc26, Sc27, Sc28, Sc29, Sc30, Sc31	Network layer	Gateway, edge gate-way, cloud, blockchain, database sensors
V1, V2, V3, V4, V5, V6, V7, V9, V10, V11, V12	Spoofing (fake sensor transmitting data), denial of service (theft of sensor), tempering (modification of input data), denial of service (spectrum jamming), denial of service (overproduction of sensor data), denial of service (sensor running out of battery).	Sc1, Sc2, Sc3, Sc4, Sc5, Sc6, Sc7, Sc8, Sc9, Sc10, Sc19, Sc20, Sc23	Physical layer	Sensors
V4, V6, V7, V8, V9, V10, V11, V12, V13, V14, V15, V16, V17, V18, V19, V20, V21, V22, V23, V24	Impersonation (fake input data representing real sensor), spoofing (LoRaWAN uses default A15-128 keys, tampering (integrity) DoS (interference), physical access (authentication is a single factor), attack to the VPN, spoofing DoS, bug exploitation in the SC code, elevation of privileges, events are validated as expected type/length, database connection parameters stored as plaintext in the container, spoofing, tampering, DoS, elevation of privilege, denial of service: spectrum jamming, native blockchain user keys (HTTPS), spoofing, DoS, modify the status of a wallet, smart contract overflow/underflow, BGP hijacking.	Sc7, Sc8, Sc9, Sc10, Sc11, Sc12, Sc13, Sc14, Sc15, S16, Sc17, Sc18, Sc21, Sc22, S24, Sc26	Network layer	Gateway, Edge Gateway, Cloud, blockchain, database

**Table 13 sensors-23-08720-t013:** Summary of issues in cloud computing and blockchain-based IoT.

Issues in Both Architectures	Cloud-Based IoT	Blockchain-Based IoT	Comments
Security	The cloud has multiple security measures but is still insecure and a number of issues have been found in recent years [41,78,79,80,81].	Blockchain depends only on its cryptographic signature [82,83], which is unique for each block[74], and, along with the validation of the consensus algorithm makes it tamper-proof [74].	Blockchain is very secure and shows no evidence of issues, but two major concerns can be found in the literature: the 51% attack and the forking issues.
Privacy	This approach has several solutions for privacy, but issues like data leaks and lack of trust exist [84,85,86,87,88].	Transparency and openness are building blocks of blockchain. Thus, privacy may also be considered an issue with this approach [89,90].	Improvements have been proposed to strengthen privacy, the private and consortium blockchain with an immutable ledger [72].
Losses and risks	There is a history of substantial financial losses and data leaks due to third-party involvement, and it is expected that this will grow with time [91,92].	Since its inception, the blockchain core algorithm has had no history of attacks that breached the security of the network [74].	Blockchain is very robust due to its consensus algorithm and hash key to protect against losses and maintain trust.
Scalability	The IPv6 protocol stack adds a huge overhead at the individual device level; address space is also a big concern for industry [77,91,93].	The overhead for the GUID is much less than IPv6; also, it provides 4.3 billion more address spaces than IPv6 [41]. However, scaling the blockchain to be as huge as the Internet is a challenge because throughput and latency will become very high [92].	The cloud approach is capable of efficiently managing a network spread over a wide geographical location.
Latency	Request and response time is very high and also depends on several factors, such as the speed of the network and geographical location [94,95]. A interesting solution introduced by Cisco is fog computing, which brings computing, communication, and processing closer to the user [90].	Mining is a heavyweight and time-intensive process when solving the mathematical puzzle (PoW) in peers over a blockchain network [72].	Both approaches have challenges with latency. An in-between solution could be obtained to overcome the latency issue e.g. local miners on the access level [72].
Energy consumption	Huge data centres are ingesting high amounts of energy, and this is increasing day by day as the number of connected devices and applications grows [91,96].	The mining process is considered to be inefficient in terms of energy consumption [77,93].	A cloud data centre has a big impact on the environment; therefore, blockchain deployed with local miners on the access level could be a answer to this problem.
Cost	This approach is a very costly in terms of bandwidth, maintenance and updates of hardware and software [41].	In terms of bandwidth consumption, maintenance, and upgrade costs, blockchain is a more effective than the cloud. However, it doubles the cost of business process execution [97].	A private distributed network with blockchain can efficiently handle common requests and responses, e.g., scheduling a washing machine, paying bills, obtaining a shopping list, etc., but heavy industrial processing could be conducted over the cloud.
Payment	This approach is very limited in the methods of payment, and the available modes of payment are rarely used [93,94].	Bitcoin is already a very popular example of digital currency [98]. Alternatively, there are several other choices for cryptocurrencies on the basis of the DLT of blockchain, including Ether, Litecoin, Nxt, Ripple, and Peercoin [99].	Undoubtedly, digital currency is the future currency; it may be Bitcoin or something else.
Flexibility	Forking with a centralised system is much easier to deal with [92].	Dealing with forks in a decentralised system is difficult. In blockchain, hard and soft forks may result in degrading the rating of a miner [83].	The cloud, due to its centralised architecture, can efficiently handle synchronising upgrades simultaneously across all nodes to deal with different types of forks.

**Table 14 sensors-23-08720-t014:** Use Case 1 scenarios’ threat actors and their properties. Threat = *f* (capability, motivation, catalyst, opportunity, threat score, method).

Use Case 1 Energy Management Sites					All Score 1–5: Low = 1 and High = 5			
	Threat	Threat Agent	Capability	Motivation	Catalyst	Opportunity	Threat Score Scenario	Method
Definition			What are their capabilities to act?	What is their motivation for acting?	What are What trigger actions could influence their motivation?	What opportunities do they have to act against the target platform?	Total score/threat factors (/5).	
Threat Scenario 1	Privileged. Configuration not secure PAM system is nonexistent.	Disgruntled employee.	High, privileged account holders have the opportunity for misconfiguration if no monitoring or auditing is in place.	Impact.	Company acting against the employee.	Knowledge of the system.		BLEmodify settingsof devices to disrupt the system.
Score	4	3	3	2	4	3	3	
Threat Scenario 2	Bluetooth traffic is not secured.	External actor.	External malicious actors will have the technical knowledge to attack the system.	Disruption of service.	Opportunistic.	Knowledge of the system.		BLE-DOSAttack to congest trafficOr drain the sensorbattery.
Score	3	2	3	3	2	2	3	
Threat Scenario 3	Bluetooth traffic is not secured, allowing theft of unencrypted data.	External/internal actor.	External malicious actors will have the technical knowledge to attack the system.	Competition.	Opportunistic.	Knowledge of the System.		Bluetooth, blue sniffing, stealing information over Bluetooth protocol, Bluetooth eavesdropping.
Score	2	2	3	2	2	2	3	
Threat Scenario 4	Bluetooth traffic is not secured, allowing the injection of false data affecting the integrity of data transmitted.	External/internal actor.	External malicious actors will have the technical knowledge to attack the system.	Bad PR.	Opportunistic.	Knowledge of the system.		Bluetooth protocol fuzzing.
Score	3	2	3	3	2	2	3	
Threat Scenario 5	The system is not hardened, allowing for overload.	External/internal actor.	External malicious actors will have the technical knowledge to attack the system.	Disruption of service.	Opportunistic.	Knowledge of the system.		IOTgateway, spamming gateway maxes outcapacity.
Score	3	2	3	3	2	4	4	
Threat Scenario 6	Weak physical security of Raspberry device and/or absence of monitoring services	External actor.	External malicious actors will have the technical knowledge to attack the system.	Disruption of service.	Opportunistic.	Knowledge of the system.		IOTgateway physical attackson the hardware.
Score	4	2	4	3	2	4	4	
Threat Scenario 7	Authentication methods are unprotected from MiM attacks.	External/internal actor.	External malicious actors will have the technical knowledge to attack the system.	Theft.	Opportunistic.	Knowledge of the system.		IOTgateway password capture for the wider system.
Score	3	2	3	3	2	3	3	
Threat Scenario 8	Zero-trust principles on-network devices not implemented, allowing for the entire system to be attacked on the loss of a single credential. Default passwords have not been changed (nonadherence or absence of password policies).	External/internal actor.	External malicious actors will have the technical knowledge to attack the system.	Theft.	Opportunistic.	Knowledge of the system.		Cisco Router password capture for the wider system.
Score	4	3	4	3	2	3	4	

**Table 15 sensors-23-08720-t015:** Use Case 2 scenarios’ threat actors and their properties. Threat = *f* (capability, motivation, catalyst, opportunity, threat score, method).

Use Case 2 Connected Sites					All Score 1–5: Low = 1 and High = 5			
	Threat	Threat Agent	Capability	Motivation	Catalyst	Opportunity	Threat Score Scenario	Method
Definition			What are their capabilities to act?	What is their motivation for acting?	What are their triggering actions that could influence their motivation?	What opportunities do they have to act against the target platform?	Total score/threat factors (/5).	
Threat Scenario 1	System data is transmitted in the clear, allowing for injection of false data.	Misconfiguration of sensors.	In-depth knowledge of the system.	Disruption of service.	Opportunistic.	Knowledge of the system.		Denial of service overproduction of sensor data.
Score	2	3	2	2		3	3	
Threat Scenario 2	The system is not configured to allow for integrity checks on data.	Disgruntled employee.	In-depth knowledge of the system.	Disruption of service.	Company decisions affecting employees.	Knowledge of the system.		Impersonation, fake in- put data representing real sensors.
Score	2	2	3	2	3	2	3	
Threat Scenario 3	Delegation of authority models not properly defined, allowing for the elevation of the privilege of applications to not adhere to IAM policies.	Misconfiguration.	In-depth knowledge of internal security policies.	Disruption of service.	Absence of automation.	Knowledge of the system.		Applications on the docker could break access control policies.
Score	3	3	4	2	4	2	3	
Threat Scenario 4	Lack of physical security and/or monitoring services.	Internal.	Knowledge of the commercial value of the systems.	Opportunistic.	Financial difficulties.	Lack of system monitoring.		Denial of service, theft ofsensor.
Score	4	3	4	3	3	2	4	
Threat Scenario 5	Lack of access controls (like MFA), leaving the VPN system open for exploitation.	Internal/external.	In-depth technical knowledge and knowledge of the system and internal policies.	Impact.	Competition /PR/disruption of the system.	Weak remote access and boundary controls.		Block chainattack on VPN.
Score	3	3	3	3	3	3	3	

**Table 16 sensors-23-08720-t016:** Comparison of i-TRACE with state-of-the-art mechanisms available in the literature.

Use Case 1 Energy Management Sites													
Ref. No	Domain	Approach	Threat Modelling Approach	Risk-Based	Protocol Based	Device Based	Best Practices	Immutability	Cyber Risks Reduction	Reduction in Cyber Risks	Level of Safety on Construction Sites	Comparison of Data between Blockchain and Digital Twin	Publishing Year
[110]	Energy	Smart Grid	Cyber-attack scenario	No	Basic	Basic	Yes	No	No	No	No	No	2012
[111]	Energy	Smart Grid	No	Yes	No	No	No	No	No	No	No	No	2015
[112]	Energy	Smart Grid	Basic	Yes	No	No	No	No	No	No	No	No	2010
[113]	Cyber-physical systems	CPS, in general, evaluates smart Grid SCA-DA	Cyber-attack scenario	Yes	Basic	SCADA	Yes	No	Yes	Yes	Basic	No	2015
[114]	No specific	Enriched model, cover DFD, attacker and security solution	DFD	Yes	No	No	No	No	Basic	Basic	No	No	2018
[115]	No specific	Course book	STRIDE	No	Yes	Yes	Yes	No	No	No	No	No	2014
[116]	WebRTC (real-time communication supply chain)	DFFD enrichment	STRIDE, DFD based	No	HTTPS	No	No	No	No	No	No	No	2018
[117]	Supply chain	Supply chain evaluation	Cyber attack, model TTP, STIX,	No	No	No	No	No	Basic	Basic	Basic	No	2019
[118]	Energy	Architecture-based by design distribution grid	STRIDE	Yes	Detailed	Detailed	Yes	No	Yes	Yes	Yes	No	2019
[119]	Wireless sensor networks real worlds’ scenarios	Threat modelling approach	Used STRIDE, DREAD, PASTA, STIKE	No	No	No	Yes	No	Yes	Yes	Yes	No	2022
[120]	Energy management	IoT pricing in IIoT	No	No	Yes	Yes	Yes	Yes	Yes	Yes	Yes	No	2020
[121]	Energy management	Energy cost deduction	Stochastic dominance (SD)	Yes	Yes	Yes	Yes	Yes	Yes	Yes	No	No	2022
	Energy management	i-TRACE	STRIDE	Yes	Yes	Yes	Yes	Yes	Yes	Yes	Yes	Yes	2023

## Data Availability

The data presented in this study are available on request from the corresponding author.

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
