# Peer review of "A Hybrid Methodology to Assess Cyber Resilience of IoT in Energy Management and Connected Sites"

_sensors, 2023, doi:10.3390/s23218720_

Round 1

Reviewer 1 Report

The data extraction indicates that the research design is suitable for the purpose of the research.  The systematic questionnaire makes it possible to collect data carefully, and the division of questions into distinct groups helps to guarantee that all aspects of the issue are covered. 

The procedures for gathering update data from IoT devices and ensuring immutability using blockchain is described. There is a need to add the smart contract algorithm with the details of functions used to add transactions into the blockchain. Also, there is a need to provide details in the real-world implementation of blockchain network. How much gas will be used for each transaction or which blockchain network will work best in this scenario?

There is a need to add graphs and charts which can be useful in representing the data and outcomes related to each KPI. This can make the data easier to understand and give a more accurate depiction of the KPIs.

In order to fully grasp the implications and effect of each KPI, provide them with real-world examples or case studies.

Keeping tabs on metrics "Quarterly" is mentioned in the KPIs. Providing a mechanism or platform where stakeholders may get real-time updates or more frequent reporting on these KPIs could prove useful.

While certain key performance indicators (KPIs) may already include information on how to measure that indicator (e.g., "Level of minimization of attack impacts"), additional information or alternative approaches to assessing that indicator could be helpful.

Think about how you might include these KPIs into your existing monitoring and management structures. This may provide for a more complete picture of the safety and efficiency of all linked and energy-management installations.

When considering the technical nature of the article's subject matter, the English language quality is quite good.  However, there is room for improvement in areas like sentence length and clarity. In order to guarantee the greatest quality of language and presentation, it may be helpful to have a professional editor or proofreader look through the content.

Author Response

We appreciate your time and efforts dedicated to providing insightful feedback on ways to strengthen our paper.  We have incorporated changes that reflect the detailed suggestions you have graciously provided. We also hope that our edits and the responses we provide below satisfactorily address all the issues and concerns you have noted.

Reveiwer-1:

S.#

Reviewer Comments

Reply/Rebuttals

Page No.

1

There is a need to add the smart contract algorithm with the details of functions used to add transactions into the blockchain. Also, there is a need to provide details in the real-world implementation of blockchain network. How much gas will be used for each transaction or which blockchain network will work best in this scenario?

Thank you for valuable suggestion of transactional algorithm. We have added a smart contract algorithm chart in the paper as Figure-5 for smart contract for transaction for energy management.

13

2

There is a need to add graphs and charts which can be useful in representing the data and outcomes related to each KPI. This can make the data easier to understand and give a more accurate depiction of the KPIs.

Thank you for this suggestion. We have included figure-1 and table-4 which represent the KPIs. For the sake of brief and concise of the paper, we avoid more figure and charts.

5,11 and 12

3

In order to fully grasp the implications and effect of each KPI, provide them with real-world examples or case studies.

We have already used the case study of Connected and Energy Management sites in the paper at section-4

8-18

4

Keeping tabs on metrics "Quarterly" is mentioned in the KPIs. Providing a mechanism or platform where stakeholders may get real-time updates or more frequent reporting on these KPIs could prove useful.

Yes “Quarterly” means quarter of month which would a rigorous regularly task. 

7-8

5

While certain key performance indicators (KPIs) may already include information on how to measure that indicator (e.g., "Level of minimization of attack impacts"), additional information or alternative approaches to assessing that indicator could be helpful.

Thank you for suggestion. KPI-6 is about the “Level of minimization of attack impacts” which will be appropriate and ample. We will try our best to explore any alternative approaches for this KPI.

N/A

6

Think about how you might include these KPIs into your existing monitoring and management structures. This may provide for a more complete picture of the safety and efficiency of all linked and energy-management installations.

Thank you. Application of these KPIs with the used cases already mentioned in the paper at section-4.

8

7

When considering the technical nature of the article's subject matter, the English language quality is quite good.  However, there is room for improvement in areas like sentence length and clarity. In order to guarantee the greatest quality of language and presentation, it may be helpful to have a professional editor or proofreader look through the content

Thank you for your appreciation and appreciation. We have worked out and revised the paper as per suggestion. We also hope that our edits and the responses will satisfactorily address all the issues and concerns you have noted.

N/A

Reviewer 2 Report

The article is devoted to solving cybersecurity issues. The topic of the article is relevant. The structure of the article does not correspond to that adopted in the MDPI for research articles (Introduction (including analysis of analogues), Models and methods, Results, Discussion, Conclusions). The level of English is acceptable. The article is easy to read. The figures in the article are of acceptable quality. The article cites 73 sources, most of which are not relevant. The References section is sloppy.

The following comments and recommendations can be formulated on the material of the article:

1. Authors have a sincere love for abbreviations. For example: "Keywords: Cyber Resilient model; blockchain Digital Twins; CNI; CSF; KRAs; KPIs; CRR; ASR; AVR; Safety and Security". Using abbreviations before they are spelled out in the text is a sign of bad taste. In addition, by submitting such a leapfrog as keywords, the authors guarantee that at least half of potential readers will not find this article.

2. The authors could not decide what they wrote - a review or a research article. Let's take a look at this work in the IoT blockchain cybersecurity space. The use of blockchain solves many of the problems associated with the Internet of Things, such as the issue of authentication and connectivity. The registration of each IoT device in a distributed ledger and the issuance or exclusion of access rights using a blockchain transaction enable all network participants to verify the legitimacy of connections and requests. As a result, connecting unauthorized devices and intercepting or spoofing data using a man-in-the-middle attack is a thing of the past. This is how Litecoin-based cloud blockchain platform Uniquid works. In addition to protecting against unauthorized connections, it provides fault tolerance for the authorization process due to the absence of a dedicated server. Another use of blockchain in IoT/IIoT is supply chain security. The registry makes it possible to track all stages of production and movement of components of a finished product, medicine or food, eliminating the possibility of theft or forgery. But these cases have little to do with cybersecurity. I ask the authors to justify what elements of scientific novelty they presented in the flock in the context of both of the above-mentioned directions.

3. Blockchain solutions for IoT are offered by various companies (for example, https://www.chronicled.com/), however, most implementations are experimental in nature. There are several reasons for such indecisiveness of customers: - risks and complexity of using new solutions; — a requirement to upgrade or replace incompatible equipment; - the need to refine existing software systems. Thus, despite expectations, securing IoT with the help of blockchain has not yet managed to gain noticeable popularity. I ask the authors to specify how the proposed solution is better than analogues in the specified metric of quality indicators. In addition, please describe plans for the actual implementation of the results.

4. Blockchain has qualities that allow it to be used for cyber defense. However, today the technology is not mature enough to move from the category of fashion novelties to the mainstream. Blockchain is excellent at ensuring the integrity of information, but does not provide significant advantages in terms of confidentiality and accessibility compared to other technologies. In addition, the implementation of a distributed ledger requires solving issues related to the organization of mining to verify transactions, as well as the development of standards, APIs and frameworks for IoT device manufacturers. All these real moments do not fit well with the authors' movements towards "Safety and security". Please justify the maturity of the proposed solution with an emphasis on frameworks and standards.

-

Author Response

We appreciate your time and efforts dedicated to providing insightful feedback on ways to strengthen our paper.  We have incorporated changes that reflect the detailed suggestions you have graciously provided. We also hope that our edits and the responses we provide below satisfactorily address all the issues and concerns you have noted.

Reveiwer-2:

S.#

Reviewer Comments

Rebuttals

Page No.

0

The article is devoted to solving cybersecurity issues. The topic of the article is relevant. The structure of the article does not correspond to that adopted in the MDPI for research articles (Introduction (including analysis of analogues), Models and methods, Results, Discussion, Conclusions). The level of English is acceptable. The article is easy to read. The figures in the article are of acceptable quality. The article cites 73 sources, most of which are not relevant. The References section is sloppy.

Thank you for your gratitude and guidance. We tried to complete this research paper in line with the MDPI standards, and only relevant references have been included.

N/A

1

Authors have a sincere love for abbreviations. For example: "Keywords: Cyber Resilient model; blockchain Digital Twins; CNI; CSF; KRAs; KPIs; CRR; ASR; AVR; Safety and Security". Using abbreviations before they are spelled out in the text is a sign of bad taste. In addition, by submitting such a leapfrog as keywords, the authors guarantee that at least half of potential readers will not find this article.

Thank you for your valuable comment. We have provided details, and full names for each abbreviation have been used in the paper. 

1-3

2

The authors could not decide what they wrote - a review or a research article. Let's take a look at this work in the IoT blockchain cybersecurity space. The use of blockchain solves many of the problems associated with the Internet of Things, such as the issue of authentication and connectivity. The registration of each IoT device in a distributed ledger and the issuance or exclusion of access rights using a blockchain transaction enable all network participants to verify the legitimacy of connections and requests. As a result, connecting unauthorized devices and intercepting or spoofing data using a man-in-the-middle attack is a thing of the past. This is how Litecoin-based cloud blockchain platform Uniquid works. In addition to protecting against unauthorized connections, it provides fault tolerance for the authorization process due to the absence of a dedicated server. Another use of blockchain in IoT/IIoT is supply chain security. The registry makes it possible to track all stages of production and movement of components of a finished product, medicine or food, eliminating the possibility of theft or forgery. But these cases have little to do with cybersecurity. I ask the authors to justify what elements of scientific novelty they presented in the flock in the context of both of the above-mentioned directions.

Thank you for showing your feedback. It is a novel research article wherein we have presented a hybrid methodology that specifically identifies the steps required (typically undertaken by those responsible for detecting, deterring, and disrupting cyber-attacks on CNI). The methodology leverages digital twins and distributed ledger technologies for our chosen i-TRACE. The proposed method protects against unauthorized connections with fault tolerance for the authorization process.

1-3

3

Blockchain solutions for IoT are offered by various companies (for example, https://www.chronicled.com/), however, most implementations are experimental in nature. There are several reasons for such indecisiveness of customers: - risks and complexity of using new solutions; — a requirement to upgrade or replace incompatible equipment; - the need to refine existing software systems. Thus, despite expectations, securing IoT with the help of blockchain has not yet managed to gain noticeable popularity. I ask the authors to specify how the proposed solution is better than analogues in the specified metric of quality indicators. In addition, please describe plans for the actual implementation of the results.

Thank you for such a valuable suggestion. As we have already stated in the abstract, the research presents a novel i-TRACE security-by-design CNI methodology that encompasses key CNI KPI’s and metrics to combat the growing vicarious nature of remote, well-planned, and well-executed cyber-attacks against CNI. Evaluations and comparison clearly demonstrate that i-TRACE has significant intrinsic advantages compared to existing ‘state-of-the-art’ mechanisms.

1-4,19

4

Blockchain has qualities that allow it to be used for cyber defense. However, today the technology is not mature enough to move from the category of fashion novelties to the mainstream. Blockchain is excellent at ensuring the integrity of information, but does not provide significant advantages in terms of confidentiality and accessibility compared to other technologies. In addition, the implementation of a distributed ledger requires solving issues related to the organization of mining to verify transactions, as well as the development of standards, APIs and frameworks for IoT device manufacturers. All these real moments do not fit well with the authors' movements towards "Safety and security". Please justify the maturity of the proposed solution with an emphasis on frameworks and standards.

Thank you for your valuable suggestion. We have addressed and highlighted the relevant modifications in the paper. “This research followed the generic ISO standard ISO31000 for risk monitoring and risk communication and CVSS scoring system.”

1

Reviewer 3 Report

I recommend a major revision based on the below points. Please, add a point-to-point response to each comment in the revision:

-          I am not convinced about the novelty of the manuscript. The novelty of the paper needs to be justified and clearly defined. It includes a clear difference between the available literature and previous works. The authors are asked to provide the limitations of the previous correlated works and then link those limitations to the current ideas and contributions of the current work.

-          The abstract section is fragile. Please re-write an abstract section, explain an obtained result and contribution, improve a proposed method, etc. Please delete unnecessary information.

-          The abstract also missed statistical information about the results.

-          The structure of the paper is vague. The paper needs to be restructured.

-          Do not add heading over heading. Instead, add a few lines related to the detail of a particular section before starting a sub-section.

-          Please avoid using the words "you," "we," or "our" in the manuscript. Please, consider using phrases like "in this study/paper/Proposed/method" or another appropriate phrasing. This applies to the entire manuscript.

-          Proofread the manuscript from a native English speaker. There are many typos and grammar mistakes. 

-          Generally, the entrance to the subject should be done more clearly and briefly. Consequently, please rewrite the introduction section and the related works section accordingly.

-          The heading should be literature review/related work.

-          At the end of the Introduction section, add the contributions clearly. See this paper for reference and citation ‘A novel deep learning-based approach for malware detection.' The mentioned contributions are not clear.

-          Related work/background/literature review should have a threat to a validity section. Add the ‘threat to a validity’ section at the start of the background section. In that section, state the search strings and databases explored to find the related work. See the below papers for references and citations 'Performance comparison and current challenges of using machine learning techniques in cybersecurity' and 'A Survey on Machine Learning Techniques for Cyber Security in the Last Decade'.

-          Summarise the literature in the form of a table.

-          The literature needs to be subdivided into multiple sub-sections.

-          Please make the Introduction and related work sections more productive using the following articles. Reading and using these articles and also cited in this article: A Review on Security Challenges in Internet of Things (IoT), A brief review of acoustic and vibration signal-based fault detection for belt conveyor idlers using machine learning models, An efficient deep learning-based skin cancer classifier for an imbalanced dataset, A Review of Content-Based and Context-Based Recommendation Systems, The Impact of Artificial Intelligence and Robotics on the Future Employment Opportunities

-          Comparison with the state-of-the-art is missed. You need to compare your method with the ground truth. 

-          How did the authors set parameters for their proposed algorithm? Please make sensitivities of these parameters to the performance of their proposed algorithm! 

-          Add the discussion related to the time complexity factor of AI models. See this paper for reference and citation 'Cyber Threat Detection Using Machine Learning Techniques: A Performance Evaluation Perspective'.

-          Some statistical tests, e.g., Wilcoxon rank-sum/signed-rank test or t-test, or ANOVA, should be implemented for the experimental results, aiming to check the significant differences among compared methods. The authors need to perform any statistical test to validate the significance of their proposed method.

-          Expand the critical results in the conclusion. Focus on the main developments in the finale. Also, write the main contributions in the conclusion.

-          Add a discussion related to adversarial attacks. See this paper: A Novel Method for Improving the Robustness of Deep Learning-based Malware Detectors against Adversarial Attacks

Overall, the paper has many inconsistencies, and the contributions are unclear. The results are not compared with the ground truth properly. Limitations are not provided in their current approach. Future directions are not clearly stated.

I am looking forward to seeing your revised version. 

All the best. 

major revision

Author Response

Thank you for your time and efforts dedicated to providing insightful feedback on ways to strengthen our paper. It is with great pleasure that we resubmit our article for further consideration. We have incorporated changes that reflect the detailed suggestions you have graciously provided. We also hope that our edits and the responses we provide below satisfactorily address all the issues and concerns you have noted.

Reveiwer-3:

S.#

Reviewer Comments

Rebuttals

page No.

1

I am not convinced about the novelty of the manuscript. The novelty of the paper needs to be justified and clearly defined. It includes a clear difference between the available literature and previous works. The authors are asked to provide the limitations of the previous correlated works and then link those limitations to the current ideas and contributions of the current work.

Thank you for showing your concerns for a better presentation of our paper. As mentioned in the paper we present i-TRACE security by designing CNI methodology that encompasses key CNI KPI’s and metrics to combat the growing vicarious nature of remote, well-planned, and well-executed cyber-attacks against CNI. It is a novel research idea compared to existing ‘state-of-the-art’ mechanisms.

1

2

The abstract section is fragile. Please re-write an abstract section, explain an obtained result and contribution, improve a proposed method, etc. Please delete unnecessary information.

Thank you for your valuable suggestion. In the revised paper, we try our best with minimum errors and omissions.  

1

3

The abstract also missed statistical information about the results.

Thank you for suggestion. Already included as highlighted.  

1

4

The structure of the paper is vague. The paper needs to be restructured.

Thank you for valuable suggestion. Needful have been done and some subtopics have been excluded.

5,7,8

5

Do not add heading over heading. Instead, add a few lines related to the detail of a particular section before starting a sub-section.

Needful have been done, as excluding more subheadings.

5,7,8

6

Please avoid using the words "you," "we," or "our" in the manuscript. Please, consider using phrases like "in this study/paper/Proposed/method" or another appropriate phrasing. This applies to the entire manuscript.

Needful have been done.

Whole paper

7

Proofread the manuscript from a native English speaker. There are many typos and grammar mistakes. 

Needful have been done

Whole paper

8

Generally, the entrance to the subject should be done more clearly and briefly. Consequently, please rewrite the introduction section and the related works section accordingly.

Needful has been done

1-3

9

The heading should be literature review/related work.

Changed as Literature Review

3

10

At the end of the Introduction section, add the contributions clearly. See this paper for reference and citation ‘A novel deep learning-based approach for malware detection.' The mentioned contributions are not clear.

Needful have been done as highlighted section.

3

11

Related work/background/literature review should have a threat to a validity section. Add the ‘threat to a validity’ section at the start of the background section. In that section, state the search strings and databases explored to find the related work. See the below papers for references and citations 'Performance comparison and current challenges of using machine learning techniques in cybersecurity' and 'A Survey on Machine Learning Techniques for Cyber Security in the Last Decade'.

Thank you for valued suggestion. We have already reduced some subtopic on the direction of a respected reviewer.

N/A

12

Summarise the literature in the form of a table.

Thank you for the valuable suggestion. We have already included more tables in the paper. Just for the sake of short and not bother for reader, we avoid to add more tables.

N/A

13

The literature needs to be subdivided into multiple sub-sections.

As already stated in the point No. 11 that we have reduced the subheading.

5,7,8

14

Please make the Introduction and related work sections more productive using the following articles. Reading and using these articles and also cited in this article: A Review on Security Challenges in Internet of Things (IoT), A brief review of acoustic and vibration signal-based fault detection for belt conveyor idlers using machine learning models, An efficient deep learning-based skin cancer classifier for an imbalanced dataset, A Review of Content-Based and Context-Based Recommendation Systems, The Impact of Artificial Intelligence and Robotics on the Future Employment Opportunities

Thank you for your observation. Improved further in the revised version.

N/A

15

Comparison with the state-of-the-art is missed. You need to compare your method with the ground truth. 

Thank you for suggestion We have already compared it with traditional methods in the highlighted sections.

1,4

16

How did the authors set parameters for their proposed algorithm? Please make sensitivities of these parameters to the performance of their proposed algorithm! 

Thank you for valued suggestion. We have already stated in the section-3 of the research paper as highlighted. 

4

17

Add the discussion related to the time complexity factor of AI models. See this paper for reference and citation 'Cyber Threat Detection Using Machine Learning Techniques: A Performance Evaluation Perspective'.

Thank you for suggestion and reference of the article. The time complexity factors are not more relates with the proposed work.

N/A

18

Some statistical tests, e.g., Wilcoxon rank-sum/signed-rank test or t-test, or ANOVA, should be implemented for the experimental results, aiming to check the significant differences among compared methods. The authors need to perform any statistical test to validate the significance of their proposed method.

Thank you for the valued suggestion. We will be used such statistical test in the next phase of the series of our work.

19

Expand the critical results in the conclusion. Focus on the main developments in the finale. Also, write the main contributions in the conclusion.

Thank you for highlighting the point. We have revised the contribution and already included the contribution in the conclusion section as highlighted in the attached paper.

3 and 19

20

Add a discussion related to adversarial attacks. See this paper: A Novel Method for Improving the Robustness of Deep Learning-based Malware Detectors against Adversarial Attacks

Thank you for the valued suggestion and reference of the related paper. As already stated in the above point. We have already reduced the paper on the suggestion and remarks by the respected reviewers and for the shorter and impressiveness of the paper. 

N/A

Reviewer 4 Report

Suggestions for Authors:

  • Provide more technical details on the digital twin, blockchain implementations, and platforms used.
  • Enhance the risk and vulnerability assessments using standard frameworks and data sources like CVSS.
  • Illustrate the KPI methodology through a detailed example calculation with sample or real data.
  • Elaborate on the selection methodology for the specific KPIs chosen and their mapping to security properties.
  • Expand the evaluation with more rigorous experiments, simulations, testbeds etc. to validate the approach. Comparisons to alternatives would also help.

Questions:

  1. What platforms and tools were used to implement the digital twins and blockchain components? Were these actual implementations or conceptual?
  2. How were the specific KPIs and their measurement formulas selected? Is there a formal methodology behind this?
  3. Can you illustrate the KPI calculation process through a detailed example with sample or real data? An end-to-end example would provide useful intuition.
  4. The risk assessments lack details compared to standard frameworks like CVSS. Can you enrich the assessments using such standards and data sources?

Author Response

Thank you for your time and efforts dedicated to providing insightful feedback on ways to strengthen our paper. It is with great pleasure that we resubmit our article for further consideration. We have incorporated changes that reflect the detailed suggestions you have graciously provided. We also hope that our edits and the responses we provide below satisfactorily address all the issues and concerns you have noted.

Reveiwer-4:

S.#

Reviewer Comments

Rebuttals

Page No.

1

Provide more technical details on the digital twin, blockchain implementations, and platforms used.

Thank you for your valuable suggestion. The next proposals will provide more details about implementing Digital Twin and Blockchain. 

N/A

2

Enhance the risk and vulnerability assessments using standard frameworks and data sources like CVSS.

Thank you for your valuable suggestion. We have highlighted on the document that we have followed the CVSS and ISO standard ISO31000 standards for risk monitoring and communication.

1-2

3

Illustrate the KPI methodology through a detailed example calculation with sample or real data.

Thank you for highlighting the issue. We have already implemented the used cases in section 4.

9-18

4

Elaborate on the selection methodology for the specific KPIs chosen and their mapping to security properties.

Thank you for your valuable suggestion. As each KPI have specified for respective performance evaluation, we have already discussed the selection and evaluation in the paper. 

4

5

Expand the evaluation with more rigorous experiments, simulations, testbeds etc. to validate the approach. Comparisons to alternatives would also help.

Thank you for your constructive ideas. We will expend more evolution experiments and simulation testbeds to validate the method in the next research phase in the future.

N/A

Question

S.#

Questions

Reply

Page No.

1

What platforms and tools were used to implement the digital twins and blockchain components? Were these actual implementations or conceptual?

Thank you for valuable suggestion. As highlighted in the paper, we have included the platform and tools to implement the Blockchain and Digital Twin.

3

2

How were the specific KPIs and their measurement formulas selected? Is there a formal methodology behind this?

Critical Success Factors (CSFs) are the key areas or activities that are critical for achieving the defined objectives. These are the high-level factors that drive success in the organisation. Identifying CSFs helps in focusing on the most important aspects of performance.

2

3

Can you illustrate the KPI calculation process through a detailed example with sample or real data? An end-to-end example would provide useful intuition.

We have already used such illustrations and examples in the paper in section 4.1as highlighted

10

4

The risk assessments lack details compared to standard frameworks like CVSS. Can you enrich the assessments using such standards and data sources?

Thank you for your valuable suggestion. We have added the risk assessments using CVSS in the paper, as highlighted.

5

Round 2

Reviewer 2 Report

I formulated the following remarks to the basic version of the article:

1. Authors have a sincere love for abbreviations. For example: "Keywords: Cyber Resilient model; blockchain Digital Twins; CNI; CSF; KRAs; KPIs; CRR; ASR; AVR; Safety and Security". Using abbreviations before they are spelled out in the text is a sign of bad taste. In addition, by submitting such a leapfrog as keywords, the authors guarantee that at least half of potential readers will not find this article.

2. The authors could not decide what they wrote - a review or a research article. Let's take a look at this work in the IoT blockchain cybersecurity space. The use of blockchain solves many of the problems associated with the Internet of Things, such as the issue of authentication and connectivity. The registration of each IoT device in a distributed ledger and the issuance or exclusion of access rights using a blockchain transaction enable all network participants to verify the legitimacy of connections and requests. As a result, connecting unauthorized devices and intercepting or spoofing data using a man-in-the-middle attack is a thing of the past. This is how Litecoin-based cloud blockchain platform Uniquid works. In addition to protecting against unauthorized connections, it provides fault tolerance for the authorization process due to the absence of a dedicated server. Another use of blockchain in IoT/IIoT is supply chain security. The registry makes it possible to track all stages of production and movement of components of a finished product, medicine or food, eliminating the possibility of theft or forgery. But these cases have little to do with cybersecurity. I ask the authors to justify what elements of scientific novelty they presented in the flock in the context of both of the above-mentioned directions.

3. Blockchain solutions for IoT are offered by various companies (for example, https://www.chronicled.com/), however, most implementations are experimental in nature. There are several reasons for such indecisiveness of customers: - risks and complexity of using new solutions; — a requirement to upgrade or replace incompatible equipment; - the need to refine existing software systems. Thus, despite expectations, securing IoT with the help of blockchain has not yet managed to gain noticeable popularity. I ask the authors to specify how the proposed solution is better than analogues in the specified metric of quality indicators. In addition, please describe plans for the actual implementation of the results.

4. Blockchain has qualities that allow it to be used for cyber defense. However, today the technology is not mature enough to move from the category of fashion novelties to the mainstream. Blockchain is excellent at ensuring the integrity of information, but does not provide significant advantages in terms of confidentiality and accessibility compared to other technologies. In addition, the implementation of a distributed ledger requires solving issues related to the organization of mining to verify transactions, as well as the development of standards, APIs and frameworks for IoT device manufacturers. All these real moments do not fit well with the authors' movements towards "Safety and security". Please justify the maturity of the proposed solution with an emphasis on frameworks and standards.

The authors answered my questions. In general, these answers suit me. I support the publication of the current version of the article. I wish the authors creative success.

Author Response

Reviewers Feedback and Rebuttals-Round 2nd

Thank you for inviting us to submit a revised draft of our manuscript, "[A Hybrid Methodology to assess Cyber Resilience of IoT in Energy Management and Connected Sites". We appreciate your time and efforts dedicated to providing insightful feedback on ways to improve and strengthen our paper. Thus, it is with great pleasure that we resubmit our article for further consideration.

We have incorporated changes that reflect the detailed suggestions you have graciously provided. We also hope that our edits and responses below satisfactorily address all the issues and concerns you and the reviewers have noted.

To facilitate your review of our revisions, the following is a point-by-point Comments and Rebuttals/Reply of each observation of reviewers are incorporated as follows:

Reviewer-2:

S.#

Reviewer Comments

Reply/Rebuttals

Page No.

1

Authors have a sincere love for abbreviations. For example: "Keywords: Cyber Resilient model; blockchain Digital Twins; CNI; CSF; KRAs; KPIs; CRR; ASR; AVR; Safety and Security". Using abbreviations before they are spelled out in the text is a sign of bad taste. In addition, by submitting such a leapfrog as keywords, the authors guarantee that at least half of potential readers will not find this article.

Thank you for your valuable feedback. We have corrected all the abbreviations in the document.

1-3

2

The authors could not decide what they wrote - a review or a research article. Let's take a look at this work in the IoT blockchain cybersecurity space. The use of blockchain solves many of the problems associated with the Internet of Things, such as the issue of authentication and connectivity. The registration of each IoT device in a distributed ledger and the issuance or exclusion of access rights using a blockchain transaction enable all network participants to verify the legitimacy of connections and requests. As a result, connecting unauthorized devices and intercepting or spoofing data using a man-in-the-middle attack is a thing of the past. This is how Litecoin-based cloud blockchain platform Uniquid works. In addition to protecting against unauthorized connections, it provides fault tolerance for the authorization process due to the absence of a dedicated server. Another use of blockchain in IoT/IIoT is supply chain security. The registry makes it possible to track all stages of production and movement of components of a finished product, medicine or food, eliminating the possibility of theft or forgery. But these cases have little to do with cybersecurity. I ask the authors to justify what elements of scientific novelty they presented in the flock in the context of both of the above-mentioned directions. Blockchain solutions for IoT are offered by various companies (for example, https://www.chronicled.com/), however, most implementations are experimental in nature. There are several reasons for such indecisiveness of customers: - risks and complexity of using new solutions; — a requirement to upgrade or replace incompatible equipment; - the need to refine existing software systems. Thus, despite expectations, securing IoT with the help of blockchain has not yet managed to gain noticeable popularity. I ask the authors to specify how the proposed solution is better than analogues in the specified metric of quality indicators. In addition, please describe plans for the actual implementation of the results.

Thank you for your valuable suggestion. The submitted paper is a novel research article wherein we have presented a hybrid methodology that specifically identifies the steps required (typically undertaken by those responsible for detecting, deterring, and disrupting cyber-attacks on CNI). The methodology leverages digital twins and distributed ledger technologies for our chosen i-TRACE. The proposed approach protects against unauthorised connections with fault tolerance for the authorisation process.

The Wilcoxon signed-rank test is used for statistical testing of the proposed approach for comparison with the state-of-the-art methods for all evaluation metrics. The p-value for all evaluation metrics is less than 0.05, which indicates that the proposed approach is better than the analogues.

1-4, 39

3

Blockchain has qualities that allow it to be used for cyber defense. However, today the technology is not mature enough to move from the category of fashion novelties to the mainstream. Blockchain is excellent at ensuring the integrity of information, but does not provide significant advantages in terms of confidentiality and accessibility compared to other technologies. In addition, the implementation of a distributed ledger requires solving issues related to the organization of mining to verify transactions, as well as the development of standards, APIs and frameworks for IoT device manufacturers. All these real moments do not fit well with the authors' movements towards "Safety and security". Please justify the maturity of the proposed solution with an emphasis on frameworks and standards.

Thank you for valuable suggestion. The observation is already addressed in the paper as highlighted that “This research followed the generic ISO standard ISO31000 for risk monitoring and risk communication and CVSS scoring system.”

2

Reviewer 3 Report

The authors have addressed my comments sufficiently. 

Congratulations

Minor editing of English language required

Author Response

Reviewers Feedback and Rebuttals-Round 2nd

Thank you for inviting us to submit a revised draft of our manuscript, "[A Hybrid Methodology to assess Cyber Resilience of IoT in Energy Management and Connected Sites". We appreciate your time and efforts dedicated to providing insightful feedback on ways to improve and strengthen our paper. Thus, it is with great pleasure that we resubmit our article for further consideration.

We have incorporated changes that reflect the detailed suggestions you have graciously provided. We also hope that our edits and responses below satisfactorily address all the issues and concerns you and the reviewers have noted.

To facilitate your review of our revisions, the following is a point-by-point Comments and Rebuttals/Reply of each observation of reviewers are incorporated as follows:

Reviewer-3:

S.#

Reviewer Comments

Rebuttals

Page No.

Minor editing of English language required.

Thank you for highlighting the minor changes. We try our best to present revised paper. 

Whole paper

Reviewer 4 Report

  1. What quantitative metrics are used to evaluate the improvements in interoperability, immutability and attack surface reduction? How are these metrics calculated?
  2. How does the methodology address real-time detection and response to cyber attacks on the IoT infrastructure? Does it integrate any intrusion detection capabilities?
  3. What are the overheads introduced by the blockchain component in terms of computation, storage and communication? How does this impact the suitability for time-critical IoT applications?

Author Response

Reviewers Feedback and Rebuttals-Round 2nd

Thank you for inviting us to submit a revised draft of our manuscript, "[A Hybrid Methodology to assess Cyber Resilience of IoT in Energy Management and Connected Sites". We appreciate your time and efforts dedicated to providing insightful feedback on ways to improve and strengthen our paper. Thus, it is with great pleasure that we resubmit our article for further consideration.

We have incorporated changes that reflect the detailed suggestions you have graciously provided. We also hope that our edits and responses below satisfactorily address all the issues and concerns you and the reviewers have noted.

To facilitate your review of our revisions, the following is a point-by-point Comments and Rebuttals/Reply of each observation of reviewers are incorporated as follows:

Reviewer-4:

S.#

Reviewer Comments

Rebuttals

Page No.

1

What quantitative metrics are used to evaluate the improvements in interoperability, immutability and attack surface reduction? How are these metrics calculated?

Thank you for highlighting this critical issue. As highlighted in the paper, we have already provided new material relevant to interoperability, immutability and attack surface methods.

8, 32,37 and 38

2

How does the methodology address real-time detection and response to cyber attacks on the IoT infrastructure? Does it integrate any intrusion detection capabilities?

Thank you for your valuable feedback. The proposed solution addresses the issue as:

  1. This research study presents an I-TRACE security by-design CNI (Critical National Infrastructure) methodology, which has key CNI-KPIs and metrics to combat the growing vicarious nature of remote, well-planned, and well-executed cyber-attacks against CNI. 
  2. The proposed solution integrates the intrusion detection capabilities as highlighted in the revised paper.  

1, 26-27

3

What are the overheads introduced by the blockchain component in terms of computation, storage and communication? How does this impact the suitability for time-critical IoT applications?

This research presents in detail the blockchain component in terms of Computation, Storage and Communication, as highlighted in the revised paper.

30-32

CONCLUDING REMARKS:

Last but not least, I would like to thank you for giving us the opportunity to improve our manuscript under the light of your valuable comments and queries. We have worked hard to incorporate your feedback and hope that these revisions persuade you to accept our submission. Thank you very much for your precious time and efforts.